# CompMarkGS: Robust Watermarking for Compressed 3D Gaussian Splatting

**Sumin In**[1]**, Youngdong Jang**[1]**, Utae Jeong**[1]**, MinHyuk Jang**[1]**, Hyeongcheol Park**[1]**,
Eunbyung Park**[2]**, Sangpil Kim**[1]*

[1]Department of Artificial Intelligence, Korea University, Republic of Korea
[2]Department of Artificial Intelligence, Yonsei University, Republic of Korea

## Abstract

As 3D Gaussian Splatting (3DGS) is increasingly adopted in various academic and commercial applications due to its high-quality and real-time rendering capabilities, the need for copyright protection is growing. At the same time, its large model size requires efficient compression for storage and transmission. However, compression techniques, especially quantization-based methods, degrade the integrity of existing 3DGS watermarking methods, thus creating the need for a novel methodology that is robust against compression. To ensure reliable watermark detection under compression, we propose a compression-tolerant 3DGS watermarking method that preserves watermark integrity and rendering quality. Our approach utilizes an anchor-based 3DGS, embedding the watermark into anchor attributes, particularly the anchor feature, to enhance security and rendering quality. We also propose a quantization distortion layer that injects quantization noise during training, preserving the watermark after quantization-based compression. Moreover, we employ a frequency-aware anchor growing strategy that enhances rendering quality by effectively identifying Gaussians in high-frequency regions, and an HSV loss to mitigate color artifacts for further rendering quality improvement. Extensive experiments demonstrate that our proposed method preserves the watermark even under compression and maintains high rendering quality.

## 1 Introduction

3D Gaussian Splatting (3DGS) (Kerbl et al., 2023) has recently emerged as an impactful method for novel view synthesis, offering real-time and photorealistic rendering capabilities. Since its introduction, 3DGS has undergone rapid advancements in the field of image-based 3D reconstruction (Charatan et al., 2024; Lin et al., 2024; Zhang et al., 2024), attracting significant attention from academia and industry. One drawback of 3DGS is that it is composed of a vast number of 3D Gaussians, requiring substantial storage space. This limitation has motivated the development of 3DGS compression techniques to reduce storage requirements and facilitate efficient transmission of 3DGS models. In parallel, its growing commercial adoption in areas such as digital twin construction and AR/VR applications has raised critical concerns regarding the copyright protection of trained 3DGS models. These concerns have led to increased research interest in 3DGS watermarking. However, to the best of our knowledge, no prior work has specifically addressed the challenge of creating a 3DGS watermark that can withstand model compression, a scenario illustrated in Fig. 1.

Existing 3DGS watermarking methods (Chen et al., 2024b; Guo et al., 2024; Huang et al., 2025; Jang et al., 2025; Tan et al., 2024) embed watermarks by directly modifying Gaussian attributes, which makes the watermark vulnerable to loss when those attributes are altered during compression. In particular, quantization-based compression shifts the distribution of model parameters during the compression process, severely degrading watermark performance. As a result, existing methods fail to provide adequate copyright protection for compressed 3DGS models.

To handle this problem, we propose CompMarkGS, an anchor-based 3DGS watermarking method. Our method leverages the anchor-based 3DGS architecture, which dynamically predicts Gaussian attributes from anchor points via multiple implicit multi-layer perceptrons (MLPs). This implicit

---

*Corresponding author.

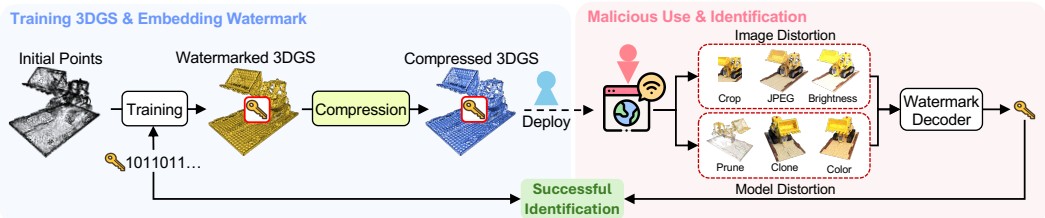

Figure 1: Application scenario of CompMarkGS. Owners embed the watermark into a 3DGS asset using our proposed method, then compress and distribute the model across various digital environments. Owners can extract the hidden message to verify ownership even if an unauthorized user alters the distributed model or its rendered images.

representation makes it extremely difficult for a malicious user to detect the presence of a watermark by directly analyzing the point cloud or Gaussian attributes. We seamlessly integrate the watermark by embedding a learnable watermark feature into the anchor feature, which is one of the anchor attributes, while keeping the rendering backbone unchanged. Unlike prior work focusing on image-domain distortions, we tackle the novel challenge of watermark robustness against model quantization. We propose a quantization distortion layer (QDL) that injects quantization noise into anchor attributes during training, teaching the watermark to be resilient. As a result, our model achieves both high rendering quality and robust watermark integrity after compression.

To further improve rendering quality, we introduce a frequency-aware anchor growing strategy. Specifically, we identify high-frequency regions in the rendered image, map them to Gaussian coordinates, and selectively apply anchor growing in those regions. In addition to this structural enhancement, we design an HSV loss to address color artifacts that occur when embedding the watermark. This loss function operates by constructing a binary mask over regions with noticeable color artifacts, identified by analyzing the hue (H), saturation (S), and value (V) components. This mask guides the loss computation by focusing on these specific regions, improving the rendering quality.

Our extensive experiments on the Blender (Mildenhall et al., 2021), LLFF (Mildenhall et al., 2019), and Mip-NeRF 360 (Barron et al., 2022) datasets show that CompMarkGS preserves watermark integrity and rendering quality more effectively both before and after compression. Each component of our method—anchor-based 3DGS watermarking, the quantization distortion layer, and the frequency-aware anchor growing strategy—independently enhances watermark fidelity while preserving rendering quality, even under compression and various watermarking attacks. Furthermore, our method outperforms state-of-the-art approaches across various watermark message lengths. Our main contributions are as follows:

- We propose CompMarkGS, a compression-tolerant anchor-based watermarking method for 3DGS that embeds a learnable watermark embedding feature into the anchor feature, preserving the model structure while ensuring high rendering quality and bit accuracy.

- We introduce a quantization distortion layer that injects quantization noise during training, enabling 3DGS watermarking to remain robust against quantization-based compression while maintaining high rendering quality.

- We propose a frequency-aware anchor growing strategy that selectively expands anchors in high-frequency regions, which enhances rendering quality, and an HSV loss that mitigates color artifacts to further improve rendering quality.

## 2 RELATED WORK

**3D Gaussian Splatting representation.** 3D Gaussian Splatting (3DGS) (Kerbl et al., 2023) is an innovative 3D representation technique that models scenes using explicit primitives. It renders images by projecting these primitives into arbitrary viewpoints and seamlessly blending pixel colors via $\alpha$-blending, achieving high-quality results while substantially accelerating real-time rendering. Thanks to these advantages, 3DGS has been widely adopted in various research areas, including avatars (Abdal et al., 2024; Chen et al., 2025a; Qian et al., 2024; Yuan et al., 2024), dynamic scenes (Li et al., 2024; Wu et al., 2024; Yan et al., 2025; Zhu et al., 2025), and 3D generation (Tang et al., 2023; Xie et al., 2024b; Yi et al., 2024; Zhou et al., 2024). Recently, Scaffold-GS (Lu et al.,

2024) has advanced the 3DGS framework by introducing anchor points to construct a hierarchical 3D representation. Anchor-based methods (Lee et al., 2024b; Ren et al., 2024; Wang et al., 2024b) effectively minimize redundant Gaussians, enhancing rendering quality and increasing robustness against view changes. This representation approach is promising for 3DGS compression optimization, as it reduces parameter counts while maintaining high-quality rendering.

**3D Gaussian Splatting compression.** To achieve high-quality rendering images, 3DGS generates a considerable number of Gaussians, leading to significant storage overhead. To address the storage overhead, vector quantization methods (Fan et al., 2025; Lee et al., 2024a; Navaneet et al., 2024; Xie et al., 2024a) have been widely explored. These methods prune Gaussians with minimal impact on rendering quality and use codebooks to encode the attributes of Gaussians compactly. More recently, anchor-based representations have gained significant attention in 3DGS (Chen et al., 2024a; Lu et al., 2024; Wang et al., 2024a). Compression methods (Chen et al., 2025b; 2024a; Wang et al., 2024a; Zhan et al., 2025) that leverage the structural relationships between anchors have shown superior performance. For example, HAC (Chen et al., 2024a) reduces spatial redundancies among anchors by utilizing hash grids for parameter quantization and enabling entropy modeling. ContextGS (Wang et al., 2024a) introduces a unified compression framework with a factorized prior, enabling entropy modeling of anchor features while leveraging hierarchical anchor relations to reduce redundancy among anchors. CAT-3DGS (Zhan et al., 2025) projects Gaussian primitives onto PCA-aligned triplanes and applies spatial autoregressive coding to capture spatial correlations, thereby enhancing entropy coding efficiency. Given their compression efficiency, anchor-based representations are especially advantageous for real-world applications.

**Steganography and digital watermarking.** Information hiding research is primarily divided into two categories: digital watermarking and steganography, which differ in their core objectives. Steganography aims to conceal information in digital assets with invisibility as the primary metric. Driven by advances in deep learning, various methods (Tancik et al., 2020; Biswal et al., 2024) have been proposed to hide information in media such as images and videos, with recent work (Dong et al., 2024; Li et al., 2023) extending into 3D scene representation. For example, GS-Hider (Zhang et al., 2025a) introduces the first steganography technique for 3D Gaussian Splatting (3DGS), and SecureGS (Zhang et al., 2025b) proposes an anchor-based steganographic framework that hides information using a private MLP. In contrast, digital watermarking embeds ownership data to protect assets and prioritizes robustness to distortions. Early watermarking research focused on the pixel domain (Van Schyndel et al., 1994; Wolfgang & Delp, 1996) or the frequency domain (Barni et al., 2001; Navas et al., 2008). Subsequently, deep learning-based methodologies (Luo et al., 2020; Zhu et al., 2018) demonstrated superior robustness against a variety of distortions. This research has recently expanded to protect radiance field models like NeRF (Mildenhall et al., 2021). WateRF (Jang et al., 2024) proposes a plug-and-play method that uses the frequency domain for robust watermarking. With the advent of 3DGS, GaussianMarker (Huang et al., 2025) embeds the watermark by estimating Gaussian uncertainties and integrating specific Gaussians. 3D-GSW (Jang et al., 2025) embeds the watermark by selectively removing Gaussians to minimize the impact on rendering quality. GuardSplat (Chen et al., 2024b) introduces an SH-aware embedding mechanism. However, these works only consider traditional distortions and do not account for real-world attacks such as model compression. In this paper, we propose a watermarking method that is robust to model compression.

## 3 PRELIMINARIES

**Scaffold-GS.** Scaffold-GS (Lu et al., 2024) clusters adjacent Gaussians using anchor points, reducing redundancy. Initial anchor points are placed at voxel centers, and the attributes for each anchor consist of an anchor feature $f \in \mathbb{R}^d$, scaling factor $l \in \mathbb{R}^3$, and $K$ learnable offsets $O \in \mathbb{R}^{K \times 3}$. Visible anchor points within the viewing frustum generate $K$ Gaussians, with their positions computed as follows:

$$\mu_k = x_a + O_k \odot l, \tag{1}$$

where $x_a \in \mathbb{R}^3$ is the anchor point position, $O_k \in \mathbb{R}^3$ denotes the $k$-th offset vector, $\mu_k$ is the generated $k$-th Gaussian position, and $\odot$ represents the element-wise product. The Gaussian attributes opacity $\alpha_i$, color $c_i$, quaternion $q_i$ and scale $s_i$ are predicted using separate MLPs:

$$\{\alpha_i, c_i, q_i, s_i\}_{i=1}^{K} = \text{MLP}(f, \delta_{av}, \vec{\mathbf{d}}_{av}), \tag{2}$$

where $\delta_{av} = \| x_a - x_v \|_2$ and $\vec{\mathbf{d}}_{av} = \frac{x_a - x_v}{\|x_a - x_v\|_2}$ denote the relative distance and viewing direction between the camera ($x_v$) and the anchor ($x_a$) positions.

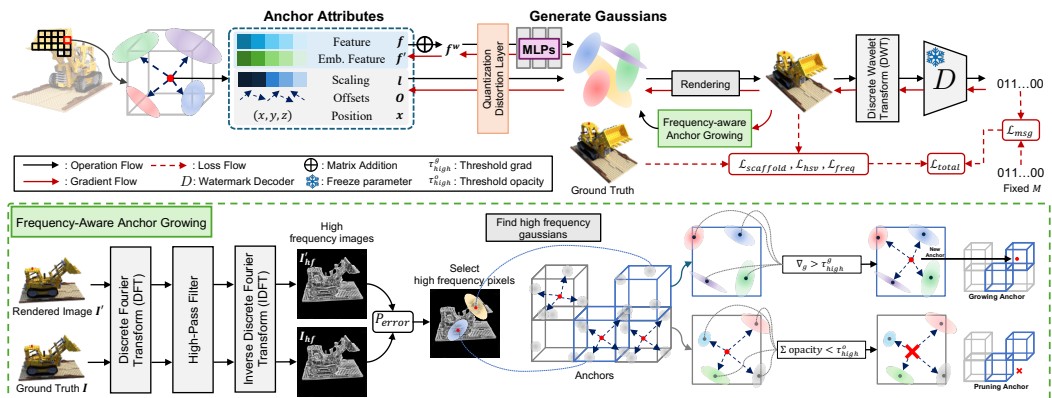

Figure 2: Overview of CompMarkGS. First, a learnable watermark embedding feature $f'$ is added to the anchor feature $f$ to create a watermarked anchor feature $f^w$. This feature is processed by the quantization distortion layer to simulate quantization noise before being fed into MLPs to predict Gaussian attributes. For watermark extraction, a pre-trained decoder retrieves the message from the low-frequency band of the rendered image. For rendering quality, our frequency-aware anchor growing strategy selectively densifies anchors in high-frequency regions. The entire framework is optimized using the total loss $\mathcal{L}_{total}$, incorporating HSV loss $\mathcal{L}_{hsv}$ and frequency loss $\mathcal{L}_{freq}$.

## 4 METHOD

### 4.1 ANCHOR-BASED 3DGS WATERMARKING

Anchor-based 3DGS (Lu et al., 2024) enhances the security of embedded watermarks by hiding explicit Gaussian attributes behind anchor features processed by implicit MLPs. This property of anchor-based 3DGS makes it more difficult for attackers to directly manipulate embedded watermarks and Gaussian attributes. Moreover, the anchor feature $f$ influences the rendering quality indirectly through MLPs rather than directly controlling Gaussian attributes. On the other hand, attributes such as position $x$, scaling factor $l$, and offsets $O$ directly determine Gaussian attributes, and modifying them leads to visible distortions in the rendered images. To embed a watermark into the anchor feature while preserving rendering quality, we introduce a learnable watermark embedding feature $f' \in \mathbb{R}^d$, with the same dimension as the anchor feature $f \in \mathbb{R}^d$. This design aims to achieve an optimal balance between rendering quality and watermark robustness. The resulting watermarked anchor feature $f^w$ is defined as follows:

$$f^w = f + \tanh(f'), \quad \text{where} \quad f, f' \in \mathbb{R}^d, \tag{3}$$

where the watermarked anchor feature $f^w$ is generated by the element-wise sum of the anchor feature $f$ and a learnable watermark embedding feature $f'$. The term $\tanh(\cdot)$ denotes the element-wise hyperbolic tangent function, which we apply to the watermark embedding feature $f'$ to bound its values within the range $[-1, 1]$ before addition. This bounding process is crucial for stabilizing gradients and preserving rendering quality. In a well-trained anchor-based 3DGS model, the components of the anchor attributes are known to follow a normal distribution (Lu et al., 2024). Adding an unbounded feature would increase the variance of this distribution, leading to unstable gradients, slower convergence, and degraded rendering quality. While a sigmoid function could be used, its $[0, 1]$ output range would introduce a directional bias by only adding positive values. The symmetric $[-1, 1]$ range of the tanh function prevents this bias, ensuring the watermark is added with a consistent magnitude. As a result, our method achieves stable convergence and efficient embedding without degrading rendering quality. A detailed analysis is provided in Appendix E.1.

### 4.2 ROBUSTNESS TO QUANTIZATION-BASED COMPRESSION

In digital watermarking research, differentiable distortion layers have long been used to enhance robustness by simulating traditional distortions, such as JPEG and cropping, during training (Zhu et al., 2018). However, this method has not been applied in existing 3DGS watermarking research to the problem of model compression, leading to significant watermark loss during quantization.

To overcome this limitation, we propose a quantization distortion layer (QDL) that simulates quantization-induced distortions, encompassing losses in watermark robustness and rendering fidelity. Our QDL leverages quantization techniques from prior anchor-based 3DGS compression methods (Chen et al., 2024a). While the original method uses this mechanism for model compression, our work is the first to adapt it as a form of data augmentation to train a robust watermark. During watermark training, the QDL simulates the quantization process by injecting noise into the watermarked anchor feature $f^w$. By integrating the QDL into the embedding stage, the watermark is thus optimized to remain robust against subsequent compression attacks. The quantized watermarked anchor feature for the $i$-th anchor $\tilde{f}_i^w$ is defined as follows:

$$\tilde{f}_i^w = f_i^w + \mathcal{U}\left(-\frac{1}{2}, \frac{1}{2}\right) \cdot q_i, \quad \text{where} \quad q_i = Q_0 \cdot \left(1 + \tanh(r_i)\right), \quad r_i = \text{MLP}_q(f_i^w), \quad (4)$$

where $q_i \in \mathbb{R}$ denotes the quantization scale restricted to $[0, 2Q_0]$, and $r_i \in \mathbb{R}$ is the refinement output from an $\text{MLP}_q$ that takes $f_i^w$ as input to adjust the initial quantization scale $Q_0$. By scaling a randomly generated $d$-dimensional noise vector from a uniform distribution over $[-\frac{1}{2}, \frac{1}{2}]$ by $q_i$ and adding it to the watermarked anchor feature $f^w$, we simulate the rounding errors caused by quantization. This process forces the watermark to become resilient to the specific distortions it will encounter during a real compression attack.

### 4.3 FREQUENCY-AWARE ANCHOR GROWING

We apply a Discrete Wavelet Transform (DWT) to the rendered image and feed its low-frequency Low-Low ($LL$) subband, $I_{LL}$, into the pre-trained HiDDeN (Zhu et al., 2018) decoder $D$ to extract the watermark message $M' = D(I_{LL})$. Therefore, we embed the watermark in the $LL$ subband to ensure robustness, while optimizing the overall rendering quality via the reconstruction loss $\mathcal{L}_{scaffold}$. However, since $\mathcal{L}_{scaffold}$ is computed on the entire image, it treats the watermark signal in the $LL$ band as an error relative to the ground truth. This creates a trade-off between the watermark loss and the reconstruction loss. In other words, improving image quality tends to weaken watermark robustness, while preserving the watermark can degrade rendering quality.

Based on this insight, we propose a frequency-aware anchor growing (FAG) strategy. To selectively grow anchors in detail-rich, high-frequency regions, our method first transforms the rendered image $I'$ and the ground truth image $I$ into the frequency domain using the Discrete Fourier Transform (DFT), as shown in Fig. 2. We then apply a high-pass filter to isolate the high-frequency components and reconstruct the corresponding high-frequency images $I'_{hf}$, and $I_{hf}$ using the Inverse Discrete Fourier Transform (IDFT). To generate a binary mask that selects pixels in these high-frequency regions, we construct an SSIM-based error map $P_{error}$. Let $N(I_{hf}, p)$ and $N(I'_{hf}, p)$ denote local $k \times k$ patches of $I_{hf}$ and $I'_{hf}$ centered at pixel $p$. We define as follows:

$$P_{error}(p) = 1 - \text{SSIM}\big(N(I_{hf}, p), N(I'_{hf}, p)\big), \quad (5)$$

where $\text{SSIM}(\cdot, \cdot)$ is standard window-based SSIM computed over patches. This SSIM-based error map is used both in the frequency loss $\mathcal{L}_{freq}$ and in FAG. After computing the median of all error values, denoted as $\tilde{P}_{error}$, we create a binary mask by selecting pixels within the range $[\tilde{P}_{error} - \epsilon, \tilde{P}_{error} + \epsilon]$. This mask is then used to extract the 2D coordinates of high-frequency regions. Subsequently, each 3D Gaussian is projected onto the 2D image plane, and its continuous coordinates are rounded to obtain integer pixel-level coordinates. By matching these 2D Gaussian coordinates with the high-frequency pixel coordinates, we generate a boolean mask $F_{mask}$ that identifies anchors located in the high-frequency regions. Only the Gaussians selected by $F_{mask}$ are activated for the anchor growing process. More implementation details are in Appendix C.4, C.5.

### 4.4 OBJECTIVE FUNCTION

**Watermark message loss.** As described in Sec.4.3, we use the pre-trained HiDDeN (Zhu et al., 2018) decoder to produce the watermark message $M'$, and the watermark message loss $\mathcal{L}_{msg}$ is defined as a binary cross entropy (BCE) loss:

$$\mathcal{L}_{msg} = -\sum_{i=1}^{L}\Big[M_i \log(\sigma(M'_i)) + (1 - M_i)\log(1 - \sigma(M'_i))\Big], \quad (6)$$

where $M \in \{0, 1\}^L$ is the ground truth message, and a sigmoid function $\sigma$ constrains $M'$ to $[0, 1]$.

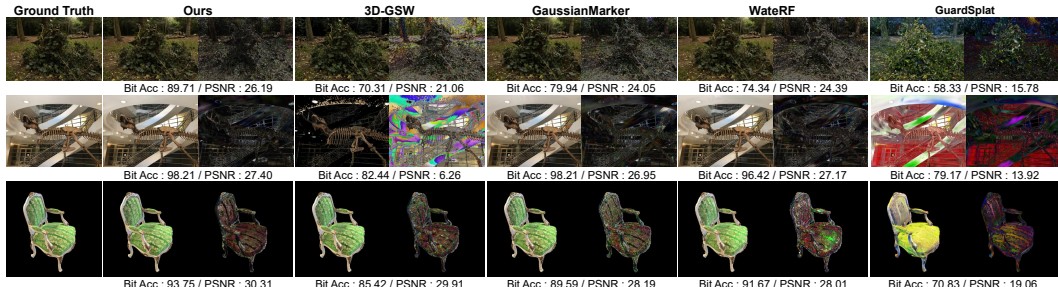

Figure 3: Qualitative comparison of rendering quality after compression between our method and baselines. For each method, the rendered image (left) and the difference map (right) are shown. Note that the difference maps are magnified five times. All results are based on 48-bit messages.

**HSV Loss.** We incorporate the HSV loss $\mathcal{L}_{hsv}$ term to minimize color artifacts during watermark embedding. Previous studies (Huang et al., 2025; Jang et al., 2024; 2025) typically computed image loss in the RGB space to improve rendering quality. However, since the RGB space calculates loss based on independent pixel-level color information, it does not accurately reflect the human visual system. Therefore, we compute image loss in the HSV space, which represents colors in a manner more similar to the human visual system. First, we construct binary masks based on hue ranges for each color to localize color artifacts. Binary mask is defined as follows:

$$M_c(p) = \begin{cases} 1, & \text{if } \text{Hue}(p) \in H_c \\ 0, & \text{otherwise} \end{cases}, \tag{7}$$

where $H_c$ denotes the hue range corresponding to color $c \in \mathcal{C}$, with $\mathcal{C} = \{R, G, B\}$. $R$, $G$, and $B$ represent the red, green, and blue color channels, respectively. The HSV loss $\mathcal{L}_{hsv}$ is defined as the mean squared error (MSE) between the pixel values of rendered image $I(p)$ and ground truth $I_{gt}(p)$:

$$\mathcal{L}_{hsv} = \frac{1}{|\mathcal{C}||\Omega|} \sum_{c \in \mathcal{C}} \sum_{p \in \Omega} \|M_c(p) \cdot (I(p) - I_{gt}(p))\|^2, \tag{8}$$

where the $|\Omega|$ is the total number of pixels, and only pixels with $M_c(p) = 1$ are penalized. The overall color loss is defined as the average of the losses for each target color in the set $\mathcal{C}$. More implementation details are in Appendix C.6.

**Total Loss.** Additionally, we employ a frequency loss $\mathcal{L}_{freq}$ to reduce high-frequency distortions during watermark embedding. The overall training loss for our method is given by:

$$\mathcal{L}_{total} = \lambda_{img}(\mathcal{L}_{scaffold} + \lambda_{hsv}\mathcal{L}_{hsv} + \lambda_{freq}\mathcal{L}_{freq}) + \lambda_{msg}\mathcal{L}_{msg}, \tag{9}$$

where $\mathcal{L}_{freq} = \frac{1}{|\mathcal{P}|} \sum_{p \in \mathcal{P}} P_{error}(p)$ is the mean of all values in the SSIM-based error map. $\mathcal{L}_{scaffold}$ represents the reconstruction loss defined in Scaffold-GS (Lu et al., 2024), which consists of one scale regularization and two rendering fidelity terms.

## 5 EXPERIMENTS

### 5.1 EXPERIMENTAL SETTINGS

**Datasets.** Following previous works (Huang et al., 2025; Jang et al., 2025), we evaluate our method using the Blender (Mildenhall et al., 2021), LLFF (Mildenhall et al., 2019), and Mip-NeRF 360 (Barron et al., 2022) datasets, comprising a total of 25 scenes. These include 8 synthetic bounded scenes from Blender, 8 forward-facing real-world scenes from LLFF, and 9 bounded real-world scenes from Mip-NeRF 360. We evaluate performance using 200 test images from Blender. For the LLFF and Mip-NeRF 360 datasets, we adopt the same data split strategy as Mip-NeRF 360.

**Implementation details.** Our method trains end-to-end on a single A6000 GPU. We conduct experiments for watermark bit-lengths of 32, 48, and 64, focusing on the 48-bit case, where we evaluate performance both before and after compression. For the decoder, we use a pre-trained HiDDeN (Zhu et al., 2018) decoder for each bit-length and keep its parameters fixed during watermark training. We use the following parameters: $\lambda_{img} = 10, \lambda_{hsv} = 0.6, \lambda_{freq} = 0.1, \lambda_{msg} = 0.45$. For the quantization scale and anchor growing range in high-frequency regions, we set $Q_0 = 1, \epsilon = 0.3$. More implementation details are in Appendix C.3.

Table 1: Quantitative comparison of bit accuracy and rendering quality before (left) and after (right) compression. Evaluations are performed using a 48-bit setting, averaged over the Blender, LLFF, and Mip-NeRF 360 datasets. Baselines are tested within an anchor-based 3DGS framework with HAC and ContextGS compression. The best results are in **bold**.

| Methods | Bit Accuracy(%) ↑ | PSNR ↑ | SSIM ↑ | LPIPS ↓ | Size(MB) ↓ |
|---|---|---|---|---|---|
| HAC + WateRF | 91.02 / 54.40 | 27.36 / 13.63 | 0.850 / 0.457 | 0.174 / 0.574 | 207.72 / 13.66 |
| HAC + GaussianMarker | 92.00 / 58.34 | 27.05 / 13.54 | 0.840 / 0.460 | 0.193 / 0.571 | 341.30 / 24.33 |
| HAC + 3D-GSW | 90.96 / 53.48 | 19.57 / 13.13 | 0.628 / 0.439 | 0.295 / 0.572 | **173.64** / 12.93 |
| HAC + GuardSplat | 79.77 / 52.72 | 16.29 / 12.28 | 0.584 / 0.425 | 0.417 / 0.609 | 215.69 / 13.15 |
| HAC + CompMarkGS | **95.95 / 95.92** | **27.68 / 27.65** | **0.856 / 0.852** | **0.171 / 0.177** | 208.96 / **12.23** |
| ContextGS + WateRF | 92.01 / 90.36 | 26.64 / 26.47 | 0.843 / 0.832 | 0.183 / 0.185 | 219.48 / 9.88 |
| ContextGS + GaussianMarker | 91.24 / 87.95 | 26.89 / 26.54 | 0.839 / 0.827 | 0.195 / 0.200 | 342.38 / 17.86 |
| ContextGS + 3D-GSW | 88.28 / 79.75 | 19.50 / 19.83 | 0.627 / 0.617 | 0.294 / 0.299 | 177.19 / 9.24 |
| ContextGS + GuardSplat | 73.20 / 67.16 | 16.90 / 16.86 | 0.653 / 0.628 | 0.353 / 0.360 | 219.82 / 9.33 |
| ContextGS + CompMarkGS | **94.36 / 94.03** | **27.60 / 27.55** | **0.845 / 0.844** | **0.172 / 0.173** | **73.39 / 5.72** |

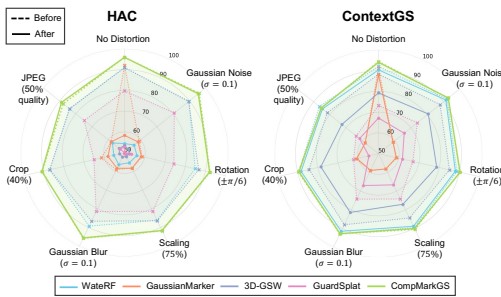

Figure 4: Robustness to image distortions. Based on 48-bit messages, the results are averaged over three datasets.

Table 2: Robustness to model distortions. Based on 48-bit messages, results are averaged over three datasets with both HAC and ContextGS before (left) and after (right) compression.

| Methods | Bit Accuracy(%) ↑ | | | |
|---|---|---|---|---|
| | No Distortion | Gaussian Noise ($\sigma$=0.005) | Clone (50%) | Prune (20%) |
| WateRF | 91.52 / 72.38 | 81.70 / 63.27 | 90.23 / 67.45 | 87.81 / 66.25 |
| GaussianMarker | 91.62 / 73.15 | 73.01 / 65.07 | 89.83 / 72.17 | 87.43 / 85.73 |
| 3D-GSW | 89.62 / 62.89 | 78.56 / 57.72 | 88.26 / 62.14 | 84.84 / 59.76 |
| GuardSplat | 76.49 / 59.94 | 65.28 / 55.53 | 73.76 / 63.20 | 69.19 / 65.19 |
| CompMarkGS | **95.16 / 94.98** | **84.58 / 80.10** | **94.35 / 94.16** | **93.53 / 93.42** |

**Baselines.** To validate our method, we evaluate watermark extraction before and after quantization-based compression in an anchor-based 3DGS framework. For comparison, we evaluate against the following four baselines: 1) WateRF (Jang et al., 2024): Embedding a watermark into NeRF (Mildenhall et al., 2021) via frequency-domain (DWT) techniques; 2) Gaussian-Marker (Huang et al., 2025): Integrating watermark-specific Gaussians into the 3D Gaussian Splatting (3DGS) (Kerbl et al., 2023) based on per-Gaussian uncertainty estimates; 3) 3D-GSW (Jang et al., 2025): Embedding the watermark message by selectively removing Gaussians with low impact on rendering quality; 4) GuardSplat (Chen et al., 2024b): Leveraging Spherical Harmonics (SH) coefficients to embed the message. All baselines originally designed for NeRF or 3DGS are adapted to the anchor-based 3DGS framework for our experiments. Notably, GuardSplat requires specific modifications due to architectural incompatibilities, which are detailed in Appendix D.1. We compare performance before and after compression using two quantization-based compression schemes: HAC (Chen et al., 2024a) and ContextGS (Wang et al., 2024a).

**Evaluation.** We evaluate the watermark performance of our proposed method, CompMarkGS, and the baselines according to three aspects: 1) Fidelity: We measure PSNR, SSIM, and LPIPS (Zhang et al., 2018) to assess the rendering quality of watermarked models before and after compression by comparing rendered images to the original images. 2) Robustness: We evaluate bit accuracy under various distortions before and after compression. We evaluate performance under distortions for rendered images, including Gaussian noise, rotation, scaling, Gaussian blur, cropping, and JPEG compression. We assess robustness against Gaussian noise, cloning, and pruning attacks for watermarked models. 3) Capacity: We measure bit accuracy for 32, 48, and 64-bit message lengths.

## 5.2 EXPERIMENTAL RESULTS

**Fidelity before and after compression.** We compare rendering quality and bit accuracy before and after compression against the baselines. The compression technique compresses anchor-based 3DGS models using quantization and entropy encoding. As shown in Tab.1, our proposed method

exhibits strong robustness to compression. While some baselines achieve excellent rendering quality and over 90% bit accuracy before compression, conventional methods not designed for compression robustness suffer from severe degradation in both watermark and scene information after compression (See Fig.3). The performance degradation is particularly noticeable in the HAC (Chen et al., 2024a) compression. HAC suffers a more significant performance drop because it uses anchor-Gaussian interpolation. This is because previous methods failed to account for errors introduced not only during quantization but also during the interpolation step, creating a compounding problem. Moreover, 3D-GSW (Jang et al., 2025) selectively removes components based on their rendering contributions before watermark training. This process eliminates both anchors and their associated Gaussians, significantly degrading rendering quality. These results confirm that our proposed watermarking method is better suited to model compression than existing methods.

**Robustness against image distortions.** We evaluate the robustness of our method against the loss of the embedded watermark when the rendered image is subjected to various distortions. We test robustness against six different image post-processing techniques: Gaussian noise ($\sigma = 0.1$), rotation (random selection within $\pm\pi/6$), scaling (75% of the original), Gaussian blur ($\sigma = 0.1$), crop (40% of the original area), and JPEG compression (50% with a quality factor of 50). As shown in Fig.4, we assess the watermark bit accuracy after post-processing and observe a decrease in bit accuracy, but the performance degradation is relatively minor compared to the baselines. These results demonstrate that our proposed watermark embedding technique is robust against image distortion.

**Robustness against model distortions.** We evaluate our method's robustness against three model distortion attacks designed to alter the embedded watermark: adding Gaussian noise ($\sigma = 0.005$) to the model parameters, randomly cloning 50% of the anchors, and randomly pruning 20% of the anchors. As shown in Tab.2, our method demonstrates superior bit accuracy and robustness compared to existing approaches under these attacks. These results confirm that our proposed method reliably preserves the watermark even when the model itself is directly manipulated. A detailed analysis is provided in Appendix E.6.

**Capacity.** Increasing the watermark message capacity leaves less embedding budget per bit and makes it more vulnerable to noise and compression, which lowers the bit accuracy. We test bit accuracy for message lengths of 32, 48, and 64 bits. As shown in Tab. 3, while the accuracy for all methods decreases with longer messages, our method maintains significantly higher bit accuracy. In contrast to other methods, the minimal performance drop even for 64-bit messages demonstrates our method's higher effective message capacity and greater robustness.

Table 3: Comparison of bit accuracy for our method and baselines at 32, 48, and 64-bit. Results are averaged over three datasets with both HAC & ContextGS before compression.

| Methods | Bit Accuracy(%) ↑ | | |
|---|---|---|---|
| | 32 bits | 48 bits | 64 bits |
| WateRF | 92.46 | 91.52 | 88.21 |
| GaussianMarker | 95.07 | 91.62 | 79.81 |
| 3D-GSW | 93.00 | 89.62 | 86.31 |
| GuardSplat | 79.64 | 76.49 | 72.83 |
| CompMarkGS | **96.52** | **95.16** | **91.29** |

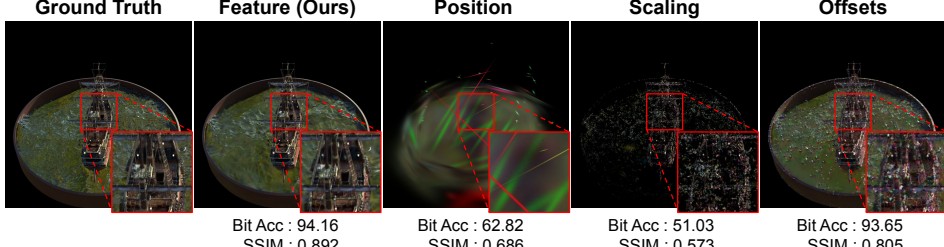

Figure 5: Qualitative comparison of rendered images with watermarks embedded into different anchor attributes. Rendered images are obtained with 48-bit watermark embedding.

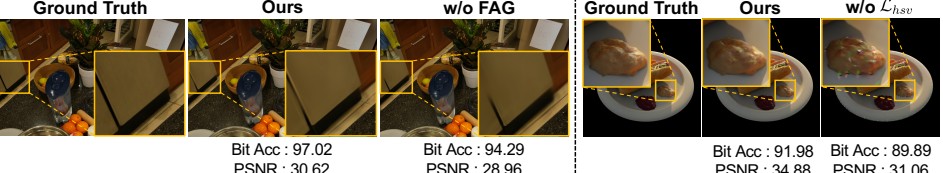

Figure 6: Qualitative comparison of rendering quality with full method (ours), without FAG (left), without $\mathcal{L}_{hsv}$ (right). Rendered images are obtained with 48-bit watermark embedding.

Table 4: Ablations of watermark embedding into different anchor attributes. Results are reported for 48-bit messages and averaged over three datasets after compression with HAC and ContextGS.

| | HAC | | | | ContextGS | | | |
|---|---|---|---|---|---|---|---|---|
| Target Parameter | Bit Acc (%)↑ | PSNR ↑ | SSIM ↑ | LPIPS ↓ | Bit Acc (%)↑ | PSNR ↑ | SSIM ↑ | LPIPS ↓ |
| Position | 67.91 | 12.05 | 0.625 | 0.350 | 82.37 | 18.49 | 0.632 | 0.432 |
| Scaling | 65.06 | 19.35 | 0.615 | 0.323 | 64.68 | 13.40 | 0.466 | 0.474 |
| Offsets | 94.88 | 24.61 | 0.792 | 0.226 | 93.98 | 25.32 | 0.755 | 0.322 |
| Anchor Feature (Ours) | **95.92** | **27.65** | **0.852** | **0.177** | **94.03** | **27.55** | **0.844** | **0.173** |

Table 5: Ablation studies on frequency-aware anchor growing (FAG), quantization distortion layer (QDL), and HSV loss $\mathcal{L}_{hsv}$. Results are reported for 48-bit messages and averaged over three datasets after compression with HAC and ContextGS.

| FAG | QDL | $\mathcal{L}_{hsv}$ | HAC | | | | ContextGS | | | |
|---|---|---|---|---|---|---|---|---|---|---|
| | | | Bit Acc(%)↑ | PSNR ↑ | SSIM ↑ | LPIPS ↓ | Bit Acc(%)↑ | PSNR ↑ | SSIM ↑ | LPIPS ↓ |
| – | – | – | 90.67 | 26.44 | 0.827 | 0.196 | 87.63 | 25.56 | 0.812 | 0.223 |
| – | ✓ | ✓ | 93.95 | 27.54 | 0.849 | 0.178 | 90.35 | 27.15 | 0.841 | 0.198 |
| ✓ | – | ✓ | 90.75 | 26.75 | 0.844 | 0.182 | 88.09 | 25.04 | 0.801 | 0.229 |
| ✓ | ✓ | – | 92.57 | 27.49 | 0.852 | 0.179 | 90.23 | 27.19 | 0.841 | 0.200 |
| ✓ | ✓ | ✓ | **95.92** | **27.65** | **0.852** | **0.177** | **94.03** | **27.55** | **0.844** | **0.173** |

## 5.3 ABLATION STUDIES

**Anchor-based 3DGS watermark embedding.** We embed the watermark into different anchor attributes (position, scaling, and offsets) and compare the rendering quality and bit accuracy. As discussed in the previous section (Sec.4.1), since the watermark embedding feature shares the same dimensions as the anchor feature, when we embed the watermark into a different attribute, we set the watermark feature to match the dimensions of that specific attribute for the experiment. As shown in Fig.5 and Tab.4, embedding the watermark into anchor attributes such as position and scaling leads to significant quality degradation, as these attributes directly adjust the Gaussian pose and shape. In contrast, our method embeds the watermark into the anchor feature, seamlessly concealing it without significantly degrading scene information.

**Effectiveness of our contributions.** We conduct an ablation study to analyze the individual contributions of our key components: frequency-aware anchor growing (FAG), quantization distortion layer (QDL), and the HSV loss $\mathcal{L}_{hsv}$. The experiments are performed under HAC and ContextGS compression settings, with the results reported in Tab.5. Removing QDL causes the most significant drop in bit accuracy, demonstrating its crucial role in preventing watermark loss during quantization. Removing FAG degrades both rendering quality and bit accuracy. This indicates that by growing anchors exclusively in high-frequency regions, FAG mitigates conflicts between low-frequency watermarks and quality enhancement algorithms, thereby securing both rendering quality and watermark robustness (See Fig.6). Finally, removing the HSV loss worsens the LPIPS score and introduces subtle color artifacts. This suggests that the HSV loss effectively mitigates color artifacts that are difficult to handle with an RGB-space loss function alone (See Fig.6). Effective watermarking must strike a balance among three competing goals: robustness, invisibility, and capacity. Our experiments show that FAG, QDL, and the HSV loss not only serve unique roles but also work complementarily to significantly improve this balance between bit accuracy and rendering quality in compressed environments.

## 6 CONCLUSION

We propose CompMarkGS, a compression-robust watermarking method. By integrating a quantization distortion layer, we train the watermark to withstand quantization-based compression, and we introduce a learnable watermark embedding feature for anchor-based insertion, achieving both high security and high fidelity. Additionally, our frequency-aware anchor growing and HSV loss preserve high-quality rendering performance. To the best of our knowledge, CompMarkGS is the first compression-robust watermarking framework for 3DGS, making it well-suited for protecting 3D assets on resource-constrained devices.

# 7 ACKNOWLEDGMENTS

This work was supported by Culture, Sports and Tourism R&D Program through the Korea Creative Content Agency grant funded by the Ministry of Culture, Sports and Tourism (International Collaborative Research and Global Talent Development for the Development of Copyright Management and Protection Technologies for Generative AI, RS-2024-00345025, 25%; Research on neural watermark technology for copyright protection of generative AI 3D content, RS-2024-00348469, 25%), the National Research Foundation of Korea(NRF) grant funded by the Korea government(MSIT)(RS-2025-00521602, 47%), Institute of Information & communications Technology Planning & Evaluation (IITP) & ITRC(Information Technology Research Center) grant funded by the Korea government(MSIT) (No.RS-2019-II190079, Artificial Intelligence Graduate School Program(Korea University), 1%; IITP-2025-RS-2024-00436857, 1%), and the Advanced GPU Utilization Support Program funded by the Government of the Republic of Korea (Ministry of Science and ICT).

## ETHICS STATEMENT

**Provenance.** We used only publicly released datasets from academic publications (Blender (Mildenhall et al., 2021), LLFF (Mildenhall et al., 2019), and Mip-NeRF 360 (Barron et al., 2022)). Additionally, the Blender and LLFF datasets are distributed under the CC BY 3.0 Unported license, while the Mip-NeRF 360 dataset is available under the Apache License 2.0.

**Anonymous.** The Blender (Mildenhall et al., 2021), LLFF (Mildenhall et al., 2019), and Mip-NeRF 360 (Barron et al., 2022) datasets represent synthetic and real-world data, respectively. The Blender dataset consists of 3D object scenes, while LLFF and Mip-NeRF 360 comprise indoor and outdoor real-world scenes. None of these datasets contains any human subjects. The details of each dataset are as shown in Tab.6.

Table 6: Dataset composition used for evaluation. Blender is synthetic, whereas LLFF and Mip-NeRF 360 are real-world scenes, with no human subjects included.

| Dataset | Scene1 | Scene2 | Scene3 | Scene4 | Scene5 | Scene6 | Scene7 | Scene8 | Scene9 |
|---|---|---|---|---|---|---|---|---|---|
| Blender | chair | drums | ficus | hotdog | lego | materials | mic | ship | – |
| LLFF | fern | flower | fortress | horns | leaves | orchids | room | trex | – |
| Mip-NeRF 360 | bicycle | bonsai | counter | flowers | garden | kitchen | room | stump | treehill |

## THE USE OF LARGE LANGUAGE MODELS

We used an LLM in a limited manner for grammar and style polishing only. It did not contribute to research ideation, experimental design, and analysis. The authors take full responsibility for all content.

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

APPENDIX

## A  BROADER IMPACTS

Our proposed CompMarkGS is an invisible watermarking method for 3D assets, such as 3D Gaussian Splatting (3DGS), which remains recoverable even after aggressive compression. Our work has three key social impacts:

- Security: CompMarkGS enables copyright protection of 3D assets distributed across diverse digital environments without compromising visual quality. This capability supports copyright protection and sustainable revenue generation across a wide range of 3D content creators.

- Applicability: CompMarkGS ensures watermark robustness under high-ratio compression, enabling protected 3DGS to be reliably utilized on resource-constrained devices, such as mobile phones and head-mounted displays. These protected 3DGS models can be safely deployed in virtual reality (VR), augmented reality (AR), medical imaging, and other 3D applications without compromising watermark integrity.

- Sustainability: To the best of our knowledge, CompMarkGS is the first watermarking method to achieve watermark robustness against compression. Building on this work, it will provide a foundation for other researchers and practitioners to extend the approach, contributing to the advancement of watermarking research across a variety of compression formats.

## B  LIMITATIONS AND FUTURE WORK

### B.1  LIMITATIONS

One limitation of our proposed method, CompMarkGS, is its reliance on a pre-trained decoder for the watermark extraction process. However, this dependency constitutes a manageable, one-time computational cost. The decoder is trained only once for a specific message length (e.g., 32, 48, or 64 bits) and can then be universally applied to any 3D Gaussian Splatting model without the need for model-specific fine-tuning. This design choice significantly enhances the method's practicality and scalability, as the initial training effort is amortized over countless applications.

### B.2  FUTURE WORK

Looking ahead, we have identified several promising directions for future work. First, we plan to extend our method to operate independently of 3D Gaussian Splatting compression, which would broaden its applicability to a wider range of scenarios. Concurrently, we aim to develop a more lightweight watermarking architecture. This initiative will focus on reducing the computational overhead of both the embedding and decoding stages, making our solution even more suitable for real-time or resource-constrained environments.

## C  MORE IMPLEMENTATION DETAILS

### C.1  TARGET COMPRESSION SCHEMES FOR ROBUSTNESS EVALUATION

To test the robustness of our watermark, we use two prominent anchor-based 3DGS compression methods, HAC (Chen et al., 2024a) and ContextGS (Wang et al., 2024a). Both techniques follow a similar pipeline, extracting features for each anchor, using MLPs to predict a context-based distribution, and then applying differentiable quantization and entropy coding. This approach significantly compresses the model size while minimizing any loss in rendering quality.

However, the information loss that occurs during quantization can damage or destroy a watermark. Therefore, our core objective is to achieve robustness specifically against this kind of quantization-based compression.

## C.2 OVERVIEW OF THE COMPMARKGS PIPELINE

In this section, we detail the core components of the CompMarkGS pipeline, covering the overall watermark training procedure.

**Watermark embedding stage.** The watermarking process begins by combining the original anchor features with a learnable watermark embedding feature to create watermarked anchor features. These features are then passed through a quantization distortion layer (QDL), which simulates distortions from quantization by injecting noise, resulting in the quantized watermarked anchor features. These final features are fed into the MLPs to predict Gaussian parameters and render the scene. To recover the watermark message from the rendered image, we apply a Discrete Wavelet Transform (DWT) to extract the low-frequency band and input it into a pre-trained decoder. The entire training process is optimized to minimize the message reconstruction loss $\mathcal{L}_{msg}$, with an additional HSV-based perceptual loss $\mathcal{L}_{hsv}$ used to prevent visual quality degradation.

**Frequency-aware anchor growing stage.** To further enhance rendering quality, we introduce the frequency-aware anchor growing (FAG) stage, which leverages frequency information to grow new anchors. We apply a Discrete Fourier Transform (DFT) and a high-pass filter to both the rendered and ground truth images to isolate their high-frequency components. A SSIM-based error map is then calculated between these two high-frequency representations. Based on this error map, we identify Gaussians in regions where high-frequency information is poorly reconstructed and use backpropagation to selectively grow new anchors in those locations.

In summary, CompMarkGS learns watermark robustness against compression-induced information loss via the QDL, while simultaneously preserving high-frequency details that could be degraded by watermarking via FAG. As a result, our proposed method achieves high watermark recovery rates and excellent visual fidelity in scenarios both with and without model compression.

## C.3 DETAILS OF TRAINING

Our proposed CompMarkGS adopts a per-scene optimization strategy and trains the anchor-based 3DGS from scratch with the watermark, in contrast to existing methods (Huang et al., 2025; Jang et al., 2024; 2025) that rely on fine-tuning pre-trained 3DGS models. For each scene, we train for $30,000$ iterations with a single $L$-bit watermark message that is kept fixed throughout training. Anchor growing is performed between $1,600$ and $15,000$ iterations, first using conventional anchor growing and then switching to the proposed frequency-aware anchor growing strategy for fine-grained refinement. The loss function for watermark training consists of the original anchor-based 3DGS (Lu et al., 2024) loss $\mathcal{L}_{scaffold}$ and three additional components: HSV loss $\mathcal{L}_{hsv}$, frequency loss $\mathcal{L}_{freq}$, and message loss $\mathcal{L}_{msg}$. Before watermark training begins, we pre-train a HiDDeN (Zhu et al., 2018) message decoder on the MS-COCO dataset (Lin et al., 2014) for 32, 48, and 64-bit messages following the strategy from 3D-GSW (Jang et al., 2025) and Stable Signature (Fernandez et al., 2023). During the subsequent watermark training, the pre-trained decoder is kept fixed. Additionally, the compression ratio of the compression method used in all experiments is set to 0.004.

## C.4 DETAILS OF THE HIGH-PASS FILTER

Early digital watermarking studies hid the watermark in the high-frequency bands of an image to keep it imperceptible. However, the watermark embedded in high-frequency regions is vulnerable to image distortions such as JPEG compression. More recent work, therefore, embeds the watermark in the low-frequency bands, which are more robust to such attacks. The drawback is that low-frequency components represent essential information, including the global structure and color distribution of the image. Modifying these components can degrade visual coherence and affect the perception of fine details such as edges and textures.

The conventional anchor growing strategy computes the mean gradient of the Gaussians within each voxel and places a new anchor point only if no anchor already exists in voxels whose mean gradient exceeds a threshold. During anchor-based 3DGS training, voxels with high mean gradients are primarily associated with regions that form the coarse structure of the scene. As a result, densification rarely occurs in fine-detail areas, and when a watermark is embedded, those under-represented regions are more susceptible to quality degradation.

To address this limitation, we propose a frequency-aware anchor growing strategy that selectively densifies Gaussians located in high-frequency regions. Specifically, we apply a Discrete Fourier Transform (DFT) to convert the rendered image into the frequency domain. To isolate high-frequency components, we use a high-pass filter mask $M_h$ that emphasizes high-frequency regions while suppressing low-frequency ones in the frequency domain:

$$M_h(p) = 1 - \exp\left(-\frac{(d(p) - \tau)^2}{2\beta}\right), \tag{10}$$

where $d(p)$ denotes the Euclidean distance between a pixel $p$ and the center of the image. $\tau$ defines the cutoff threshold of the mask and $\beta$ determines the degree of attenuation. The resulting mask $M_h$ is multiplied element-wise with the Fourier-transformed image to suppress low-frequency components and preserve only the high-frequency details. Afterward, we apply the Inverse Discrete Fourier Transform (IDFT) to reconstruct the filtered images in the spatial domain, resulting in the high-frequency rendered images $I'_{hf}$ and the high-frequency ground truth images $I_{hf}$. Further details of the frequency-aware anchor growing process can be found in the main paper (See main paper Sec.4.3).

## C.5 Details of Frequency-aware Anchor Growing

We propose a frequency-aware anchor growing strategy that efficiently identifies and densifies Gaussians in high-frequency regions to mitigate visual quality degradation caused by watermark embedding. Specifically, we first compute a SSIM-based error map $P_{error}$ (See main paper Sec.4.3) between the high-frequency rendered image $I'_{hf}$ and the ground truth image $I_{hf}$, both extracted in the frequency domain (See Sec.C.4). This error map $P_{error}$ highlights pixel regions where high-frequency information is not faithfully reconstructed. To identify pixel regions that contain visually important fine details, we compute the median value $\tilde{P}_{error}$ of the SSIM-based error map $P_{error}$. When relying on the maximum value of the SSIM-based error map $P_{error}$, a few outlier pixels can dominate the selection of high-frequency regions. To mitigate this, we instead use the median value $\tilde{P}_{error}$, which provides a more meaningful identification of high-frequency regions. We then generate a binary mask $I_{mask}$ by filtering pixels whose error values fall within the interval $[\tilde{P}_{error} - \epsilon, \tilde{P}_{error} + \epsilon]$:

$$I_{mask}(p) = \begin{cases} 1, & \text{if } |P_{error}(p) - \tilde{P}_{error}| \leq \epsilon \\ 0, & \text{otherwise} \end{cases}, \tag{11}$$

where $p$ denotes the pixel coordinates in the image. Subsequently, we construct a binary mask $G_{mask}$ to select the 3D Gaussian points that 2D projections fall within the boundaries of the image:

$$G_{mask}(k) = \begin{cases} 1, & \text{if } 0 \leq x < W \quad \text{and} \quad 0 \leq y < H \\ 0, & \text{otherwise} \end{cases}, \tag{12}$$

where $k = (x, y)$ denotes the pixel coordinates of the 2D Gaussian, and $W$ and $H$ represent the width and height of the image, respectively. Using the two binary masks $I_{mask}$ and $G_{mask}$, we generate a boolean mask $F_{mask}$ that identifies Gaussians located in specific high-frequency regions by matching their 2D coordinates to the high-frequency pixel locations:

$$F_{mask}(k) = I_{mask}(k) \cdot G_{mask}(k), \tag{13}$$

where $k$ denotes the pixel coordinates of the 2D Gaussian. During the anchor growing process, we calculate the average of Gaussians within the voxel regions selected by the computed boolean mask $F_{mask}$, based on the accumulated sum of their gradients. If the average gradient exceeds the threshold $\tau^g_{high} = 0.00015$ and no anchor exists in the corresponding voxel, a new anchor is added. The threshold for the accumulated opacity used in the subsequent anchor pruning step is set to $\tau^o_{high} = 0.15$.

## C.6 Details of HSV loss

In the field of digital watermarking, achieving imperceptible watermark embedding is a primary research goal. However, there exists an inherent trade-off between invisibility and bit accuracy.

When training a model to embed a watermark in 3D Gaussian Splatting (3DGS), color artifacts emerge due to the tendency to embed the watermark into specific Gaussians to achieve high bit accuracy. Additionally, these artifacts occur particularly in regions rendered by a small number of Gaussians, where even slight modifications to the Gaussians can lead to noticeable distortions.

Although the RGB color space is widely used for image representation, prior work on image sharpening (Kau & Lee, 2013) indicates that the RGB color space does not fully account for the perceptual characteristics of the human visual system. In particular, the human visual system is more sensitive to luminance variations than to chromatic changes (Gonzalez, 2009). Unlike the RGB color space, the Hue, Saturation, and Value (HSV) space separates chromatic information (Hue and Saturation) from luminance (Value), enabling more precise detection and suppression of color artifacts. Based on this property, we introduce an HSV loss that enhances rendering quality while preserving high bit accuracy.

To isolate specific regions in the HSV space, we construct binary masks (See main paper Eq.7) by filtering pixels based on predefined hue ranges $H_c$, along with threshold conditions on saturation $\tau_c^s$ and value $\tau_c^v$, corresponding to color $c \in \mathcal{C}$, where $\mathcal{C} = \{R, G, B\}$. Each pixel $p \in \Omega$, where $\Omega \subset \mathbb{R}^2$ denotes the spatial domain of the image, is evaluated based on its HSV values. Let $S(p) \in [0, 1]$ and $V(p) \in [0, 1]$ denote the saturation and value components at pixel $p$, respectively. We define the hue range $H_c$ for each color as follows:

$$
\begin{aligned}
H_R &\in \left[0, \frac{\pi}{3}\right) \cup \left[\frac{5\pi}{3}, 2\pi\right), \quad \text{if } S(p) \geq \tau_R^s \text{ and } V(p) \geq \tau_R^v, \\
H_G &\in \left[\frac{\pi}{3}, \pi\right), \quad \text{if } S(p) \geq \tau_G^s \text{ and } V(p) \geq \tau_G^v, \\
H_B &\in \left[\pi, \frac{5\pi}{3}\right), \quad \text{if } S(p) \geq \tau_B^s \text{ and } V(p) \geq \tau_B^v,
\end{aligned}
\tag{14}
$$

where $H(p) \in [0, 2\pi)$ denotes the hue value at pixel $p$. Pixels that do not satisfy the corresponding saturation or value thresholds are excluded from the mask, even if their hue lies within the specified range. We set thresholds for the red with $\tau_R^s = 0.4$ and $\tau_R^v = 0$, while using relaxed thresholds $\tau_c^s = 0.2, \tau_c^v = 0.2$ for the green and blue.

## D DISCUSSION

### D.1 COMPARISON WITH RELATED WORK

**Distinction from anchor-based approaches.** Although both CompMarkGS and SecureGS use an anchor-based structure, they are fundamentally different, starting from their problem definition. SecureGS (Zhang et al., 2025b) is a steganography method aiming to hide large amounts of data, with invisibility as its key metric. In contrast, CompMarkGS is a digital watermarking method for proving ownership, which prioritizes robustness. Notably, the "robustness" evaluation in SecureGS focuses on preserving the rendering quality of the hidden scene, whereas CompMarkGS aims for the recovery of the hidden message itself under attacks like model compression. To achieve these different goals, their information embedding mechanisms are also distinct. SecureGS predicts hidden Gaussian offsets with a private MLP, while CompMarkGS directly injects a learnable watermark embedding feature into the anchor feature and extracts the message from the rendered image via a pre-trained decoder. Therefore, CompMarkGS is not an incremental improvement on SecureGS, but a unique solution independently designed for the new objective of verifiable ownership after model compression.

**Comparison with 3DGS watermarking techniques.** A key difference between our method and existing watermarking methods, particularly frequency-based approaches such as 3D-GSW (Jang et al., 2025), lies in our unique approach to utilizing frequency information. While both methods leverage frequency data for Gaussian densification, their core strategies and objectives are fundamentally different. 3D-GSW measures the frequency intensity of patches, whereas CompMarkGS precisely identifies high-frequency error regions between the rendered image and the ground truth, selectively densifying anchors in those specific areas. However, the most significant distinction is the role of compression in the watermarking pipeline. Unlike 3D-GSW, which incorporates model pruning as part of its watermarking process, CompMarkGS is the first to define and solve

the novel problem of watermark robustness against external quantization compression applied to a fully trained model. To achieve this, we introduce a quantization distortion layer (QDL) that simulates quantization noise during training, a core component absent in 3D-GSW. Our contribution is proposing a unique method specifically optimized to address the previously unexplored challenge of watermark loss during quantization compression. This focus on compression robustness also sets CompMarkGS apart from GaussianMarker (Huang et al., 2025) and GuardSplat (Chen et al., 2024b). These methods address traditional image distortions like Gaussian noise and JPEG compression, or model distortions such as pruning. However, they lack a mechanism to mitigate the loss of watermark information caused by quantization compression.

Furthermore, for a fair comparison, we adapt the existing vanilla 3DGS-based watermarking techniques to the anchor-based 3DGS framework. However, during this process, GuardSplat requires significant adaptation due to a core architectural incompatibility. This is because GuardSplat relies on adding offsets to explicit Spherical Harmonics (SH) parameters, which do not exist in our anchor-based model. Therefore, we modified its mechanism to apply offsets to the anchor features instead for our experiments.

### D.2 MOTIVATION FOR WATERMARKING ANCHOR-BASED 3D GAUSSIAN SPLATTING

As the demand for lightweight and efficiently rendered 3D Gaussian Splatting (3DGS) (Kerbl et al., 2023) models grows, anchor-based architectures are emerging as a key solution. In particular, models like Scaffold-GS (Lu et al., 2024), which combine anchors with MLPs, achieve both high memory efficiency and excellent rendering quality compared to vanilla 3DGS. This anchor-based structure is highly practical, making it ideal for mobile and AR/VR environments where real-time rendering is essential.

We identified these anchor-based 3DGS models as ideal targets for watermarking due to a significant security advantage. Instead of directly storing and rendering Gaussian attributes, anchor-based models generate them indirectly by feeding anchor features into an MLP. If we embed a watermark into these anchor features, it undergoes a nonlinear transformation by the MLP. This process expresses the watermark within the Gaussian attributes in a complex and concealed manner, making it extremely difficult for an external attacker to directly interpret or remove it.

Therefore, selecting an anchor-based 3DGS model for watermarking is a rational approach that secures both practical benefits, such as a lightweight structure, and robust security through the MLP.

## E ADDITIONAL EXPERIMENTAL RESULTS

### E.1 EFFECTIVENESS OF WATERMARK EMBEDDING FEATURE

To verify the effectiveness of our proposed watermark embedding strategy, we compare the performance of two approaches: using only the anchor feature and using a dedicated watermark embedding feature. As described in Sec.4.1, our method introduces a dedicated watermark embedding feature that shares the same dimension as the anchor feature, combining them via element-wise sum. This design ensures that the total feature dimension remains constant, regardless of whether a watermark is embedded. Therefore, any observed performance increase can be attributed to the effectiveness of the embedding strategy itself, rather than an increase in model capacity.

Table 7: Ablation study on the dedicated watermark embedding feature. Results are reported for 48-bit messages and averaged over three datasets after ContextGS compression.

| Embedding Method | Bit Acc. (%) ↑ | PSNR ↑ | SSIM ↑ | LPIPS ↓ |
|---|---|---|---|---|
| Anchor Feature (Ours w/o $f'$) | 92.94 | 26.79 | 0.834 | 0.209 |
| Anchor Feature (Ours w/ $f'$) | 94.03 | 27.55 | 0.844 | 0.173 |

As shown in Tab.7, using only the anchor feature creates a conflict between the two tasks of representing the scene and hiding information, which leads to degraded rendering quality. In contrast, our approach of introducing a dedicated watermark embedding feature preserves high rendering quality while enabling a more robust watermark. We conclude that using an independent watermark em-

bedding feature is essential for effectively decoupling these two objectives, thereby achieving an optimal balance between rendering quality and watermark robustness.

## E.2 EFFECTIVENESS OF HSV LOSS

When only RGB-based loss functions are used for watermark embedding, the optimization of 3DGS fails to consider the characteristics of the human visual system. As a result, 3DGS with a watermark renders color artifacts in the rendered images, particularly in regions with strong chromatic and luminance variations. These artifacts are observed in Fig.7 and Fig.8, despite achieving high bit accuracy. To address this, we introduce an HSV loss that aligns more closely with the perceptual characteristics of the human visual system. The HSV color space separates chromatic components (Hue and Saturation) from luminance (Value), making it more suitable for detecting and reducing color artifacts. As shown in Fig. 7, the binary masks of HSV loss successfully detect color artifacts in each color channel. Moreover, Fig. 8 shows that the HSV loss allows 3DGS to focus its optimization on minimizing color artifacts, maintaining high bit accuracy. These results show that HSV loss effectively reduces color artifacts while preserving high bit accuracy and rendering quality.

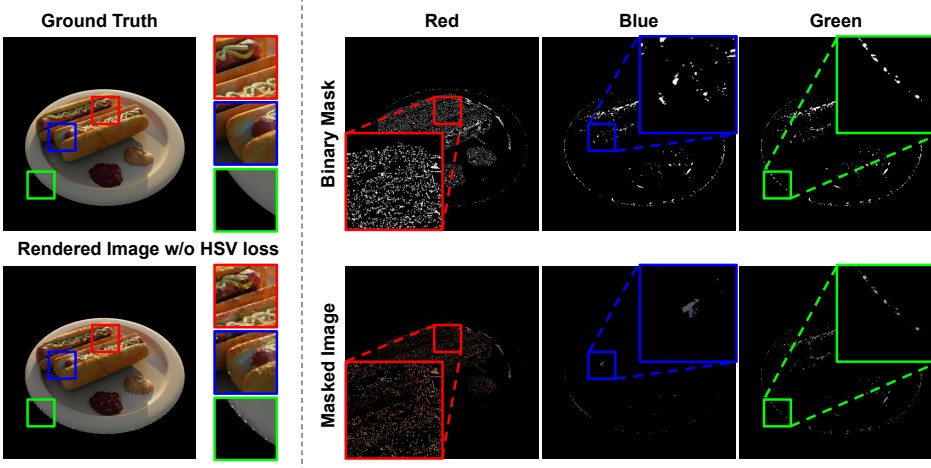

Figure 7: Visualization of the binary mask and masked image for each color channel (Red, Green, and Blue). The left column shows the ground truth and the rendered image without the HSV loss. The right columns present the binary masks (top) and masked results (bottom) for each color channel. This result is based on 48-bit messages.

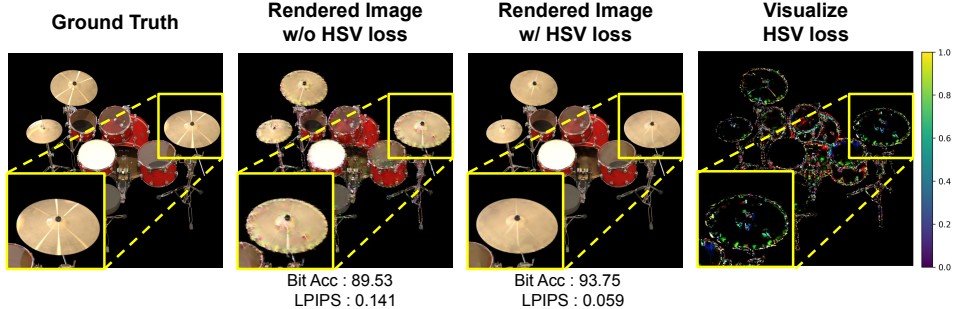

Figure 8: Visualization of HSV loss. The comparison is performed between images rendered with and without the HSV loss. Results are based on 48-bit messages.

## E.3 ROBUSTNESS UNDER VARIOUS CONDITIONS

**Robustness to pruning-based compression.** To further validate our method's robustness to model compression, we conduct an experiment comparing its watermark performance against a representative pruning-based technique, LightGaussian (Fan et al., 2025). On the Mip-NeRF 360 (Barron et al., 2022) dataset, we apply 66% pruning to the outputs of models that were already compressed with HAC (Chen et al., 2024a) and ContextGS (Wang et al., 2024a). Tab.8 compares the bit accuracy and model size before and after this pruning. The results show that CompMarkGS experiences

a significantly smaller drop in bit accuracy after pruning compared to the baselines. This is because CompMarkGS directly integrates both pruning and anchor growing into its training process, which builds resilience to the distortions caused by parameter removal. These findings corroborate the results from Fig.9, where our method maintained over 90% bit accuracy under 50% random pruning, and collectively demonstrate that our technique is highly robust to pruning-induced distortions.

Table 8: Robustness comparison against pruning-based compression using LightGaussian. Results are for the Mip-NeRF 360 dataset (48-bit messages), showing bit accuracy and model size before and after 66% pruning.

| Method | Bit Acc. (%) ↑ | Size (MB) ↓ |
|---|---|---|
| WateRF | 92.65 / 67.29 | 248.97 / 84.77 |
| GaussianMarker | 92.89 / 66.67 | 418.66 / 141.84 |
| 3D-GSW | 91.93 / 67.06 | 228.62 / 77.46 |
| CompMarkGS | **96.09 / 83.50** | **186.71 / 61.61** |

**Robustness to general quantization.** To validate the robustness of our proposed quantization distortion layer (QDL) beyond 3D Gaussian Splatting (3DGS) specific compression, we conduct an additional experiment in a general quantization environment. We use the PyTorch quantization API (Paszke et al., 2019) to quantize the model parameters from float32 to int8 and evaluate the performance on the Mip-NeRF 360 (Barron et al., 2022) dataset with a 48-bit setting. As shown in Tab.9, the results demonstrate that while existing methods suffer an average drop in bit accuracy of about 27%, CompMarkGS with QDL applied shows only about a 5% decrease, proving its superior robustness. These findings are consistent with the stable performance of CompMarkGS compared to the 22% average performance drop of existing methods shown in the main paper's Tab.1. They also align with the results in Tab.5, where applying QDL improves bit accuracy by more than 5% on average. Therefore, we have comprehensively confirmed that QDL is a key component that effectively ensures watermark robustness, not only in 3DGS-specific compression but also across diverse quantization scenarios.

Table 9: Robustness comparison under general quantization. Results are reported on the Mip-NeRF 360 dataset with 48-bit messages after quantizing models from float32 to int8.

| Method | Bit Acc. (%)↑ | PSNR ↑ | SSIM ↑ | LPIPS ↓ |
|---|---|---|---|---|
| WateRF | 93.20 / 69.20 | 26.09 / 12.17 | **0.788** / 0.135 | 0.260 / 0.610 |
| GaussianMarker | 92.49 / 66.02 | 26.33 / 15.83 | 0.775 / 0.407 | 0.282 / 0.588 |
| 3D-GSW | 91.08 / 66.20 | 18.53 / 14.93 | 0.667 / 0.374 | 0.322 / 0.662 |
| CompMarkGS | **95.59 / 90.11** | **26.56 / 21.73** | 0.777 / **0.700** | **0.248 / 0.336** |

## E.4 GENERALITY AND COMPATIBILITY

**Robustness to adversarial attacks.** In addition to the rendered image distortions and 3D Gaussian Splatting (3DGS) model attacks discussed in this paper, we evaluate our method's robustness against a more sophisticated, gradient-based adversarial attack. We performed a projected gradient descent (PGD) (Madry et al., 2017) attack specifically targeting the HiDDeN (Zhu et al., 2018) decoder. As shown in Tab.10, the bit accuracy of existing methods drops to the 50% range after the PGD attack, whereas CompMarkGS maintains a high accuracy of 84.33%. This result demonstrates that our method is highly robust not only to general distortions but also to sophisticated removal attacks, such as a PGD attack that directly targets the decoder.

**Decoder independence.** To demonstrate that our method's robustness is not dependent on a specific decoder architecture, we evaluate its performance by replacing the original HiDDeN (Zhu et al., 2018) decoder with a DINO-based SSL (Fernandez et al., 2022) decoder. The experiment measures the average performance across three datasets (Blender (Mildenhall et al., 2021), LLFF (Mildenhall et al., 2019), and Mip-NeRF 360 (Barron et al., 2022)) using 48-bit messages, comparing the results before and after HAC (Chen et al., 2024a) and ContextGS (Wang et al., 2024a) compression.

Table 10: Robustness comparison against PGD adversarial attacks. Results are for 48-bit messages, showing bit accuracy before and after the attack targeting the HiDDeN decoder.

| Method | Bit Acc. (None)↑ | Bit Acc. (PGD $\epsilon = 0.1$) ↑ |
|---|---|---|
| WateRF | 92.01 | 56.06 |
| GaussianMarker | 87.95 | 57.53 |
| 3D-GSW | 79.25 | 57.56 |
| CompMarkGS | **94.03** | **84.33** |

As shown in Tab.11, despite using the SSL decoder, the drop in bit accuracy from compression is negligible at just -0.04% for HAC and -0.09% for ContextGS, indicating highly stable performance. Furthermore, our method outperforms all baselines both before and after compression. Specifically, compared to the strongest baseline, GaussianMarker (Huang et al., 2025), our post-compression bit accuracy is 35.16% higher under HAC and 5.05% higher under ContextGS. These results confirm that CompMarkGS possesses truly decoder-agnostic characteristics.

Table 11: Performance comparison with an alternative decoder (DINO-based SSL). Results are for 48-bit messages, averaged over three datasets, showing performance before and after HAC and ContextGS compression.

| Methods | Bit Accuracy (%) ↑ | PSNR ↑ | SSIM ↑ | LPIPS ↓ |
|---|---|---|---|---|
| HAC + WateRF | 91.02 / 54.40 | 27.36 / 13.63 | 0.850 / 0.457 | 0.174 / 0.574 |
| HAC + GaussianMarker | 92.00 / 58.34 | 27.05 / 13.54 | 0.840 / 0.460 | 0.193 / 0.571 |
| HAC + 3D-GSW | 90.96 / 53.48 | 19.57 / 13.13 | 0.628 / 0.439 | 0.295 / 0.572 |
| HAC + GuardSplat | 79.77 / 52.72 | 16.29 / 12.28 | 0.584 / 0.425 | 0.417 / 0.609 |
| HAC + CompMarkGS (SSL) | 93.54 / 93.50 | 27.21 / 27.16 | 0.847 / 0.842 | 0.190 / 0.192 |
| HAC + CompMarkGS (HiDDeN) | **95.95 / 95.92** | **27.68 / 27.65** | **0.856 / 0.852** | **0.171 / 0.177** |
| ContextGS + WateRF | 92.01 / 90.36 | 26.64 / 26.47 | 0.843 / 0.832 | 0.183 / 0.185 |
| ContextGS + GaussianMarker | 91.24 / 87.95 | 26.89 / 26.54 | 0.839 / 0.827 | 0.195 / 0.200 |
| ContextGS + 3D-GSW | 88.28 / 79.75 | 19.50 / 19.83 | 0.627 / 0.617 | 0.294 / 0.299 |
| ContextGS + GuardSplat | 73.20 / 67.16 | 16.90 / 16.86 | 0.653 / 0.628 | 0.353 / 0.360 |
| ContextGS + CompMarkGS (SSL) | 93.09 / 93.00 | 25.86 / 25.82 | 0.829 / 0.828 | 0.220 / 0.221 |
| ContextGS + CompMarkGS (HiDDeN) | **94.36 / 94.03** | **27.60 / 27.55** | **0.845 / 0.844** | **0.172 / 0.173** |

### E.5 COMPARISON OF ROBUSTNESS TO IMAGE DISTORTIONS

This section provides the detailed quantitative results for the experiment on robustness to image distortions, which is visually summarized in Fig.4 of the main paper. We evaluate watermark resilience by applying six post-processing distortions to the rendered images: Gaussian noise, rotation, scaling, Gaussian blur, crop, and JPEG compression. Tab.12 details the bit accuracy for each distortion condition, both before (left) and after (right) compression. As the results show, our proposed method, CompMarkGS, consistently exhibits superior robustness to existing methods across most distortion and compression scenarios.

### E.6 COMPARISON OF ROBUSTNESS TO MODEL DISTORTIONS

Fig.9a and Fig.9b show the bit accuracy under different levels of model distortion. We conduct three model distortions: 1) adding Gaussian noise to all model parameters, 2) randomly removing anchors, and 3) randomly cloning anchors. Across different distortion strengths, our proposed method consistently outperforms the baseline. Notably, under prune and clone distortion settings, our method exhibits less performance degradation, owing to the incorporation of both pruning and anchor growing during training in CompMarkGS. These results demonstrate that our method achieves superior performance over the baselines, even under severe model distortions.

Table 12: Robustness to image distortions before (left) and after (right) compression. Evaluations are performed under the 48-bit setting, averaged over the Blender, LLFF, and Mip-NeRF 360 datasets. Baselines are tested within an anchor-based 3DGS framework using HAC and ContextGS compression. The best results are in **bold**.

| Methods | Bit Accuracy(%) ↑ | | | | | | |
|---|---|---|---|---|---|---|---|
| | No Distortion | Gaussian Noise ($\sigma = 0.1$) | Rotation ($\pm \pi/6$) | Scaling (75%) | Gaussian Blur ($\sigma = 0.1$) | Crop (40%) | JPEG Compression (50% quality) |
| HAC + WateRF | 91.02 / 54.40 | 89.75 / 55.90 | 84.92 / 56.16 | 86.17 / 55.38 | 89.33 / 56.34 | 87.00 / 55.30 | 84.00 / 57.47 |
| HAC + GaussianMarker | 92.00 / 58.34 | 59.08 / 58.33 | 58.75 / 58.33 | 58.33 / 58.33 | 59.08 / 58.33 | 61.42 / 58.17 | 58.25 / 58.33 |
| HAC + 3D-GSW | 90.96 / 53.48 | 89.50 / 52.50 | 86.67 / 52.25 | 86.42 / 52.08 | 86.67 / 52.25 | 86.92 / 52.50 | 83.83 / 53.25 |
| HAC + GuardSplat | 79.77 / 52.72 | 80.67 / 50.17 | 74.42 / 53.33 | 81.33 / 50.75 | 81.50 / 50.33 | 65.08 / 50.00 | 74.83 / 52.67 |
| HAC + CompMarkGS | **95.95 / 95.92** | **96.00 / 95.50** | **92.25 / 92.08** | **91.83 / 91.67** | **95.83 / 95.58** | **90.67 / 90.83** | **87.75 / 88.83** |
| ContextGS + WateRF | 92.01 / 90.36 | 92.25 / 91.50 | 88.75 / 88.00 | 89.58 / 88.83 | 92.75 / 91.50 | 89.50 / 88.25 | **86.41** / 84.58 |
| ContextGS + GaussianMarker | 91.24 / 87.95 | 59.08 / 58.92 | 59.00 / 58.75 | 58.25 / 58.17 | 59.17 / 58.92 | 61.08 / 60.67 | 58.25 / 58.25 |
| ContextGS + 3D-GSW | 88.28 / 79.25 | 87.92 / 80.67 | 85.08 / 78.75 | 84.42 / 77.08 | 88.00 / 81.42 | 84.33 / 78.50 | 81.58 / 72.58 |
| ContextGS + GuardSplat | 73.20 / 67.16 | 73.92 / 65.83 | 67.00 / 61.83 | 74.25 / 66.67 | 74.33 / 67.08 | 62.33 / 54.75 | 63.75 / 61.67 |
| ContextGS + CompMarkGS | **94.36 / 94.03** | **93.04 / 92.98** | **90.17 / 90.33** | **90.46 / 90.06** | **92.96 / 92.81** | **89.63 / 89.29** | 85.21 / **85.31** |

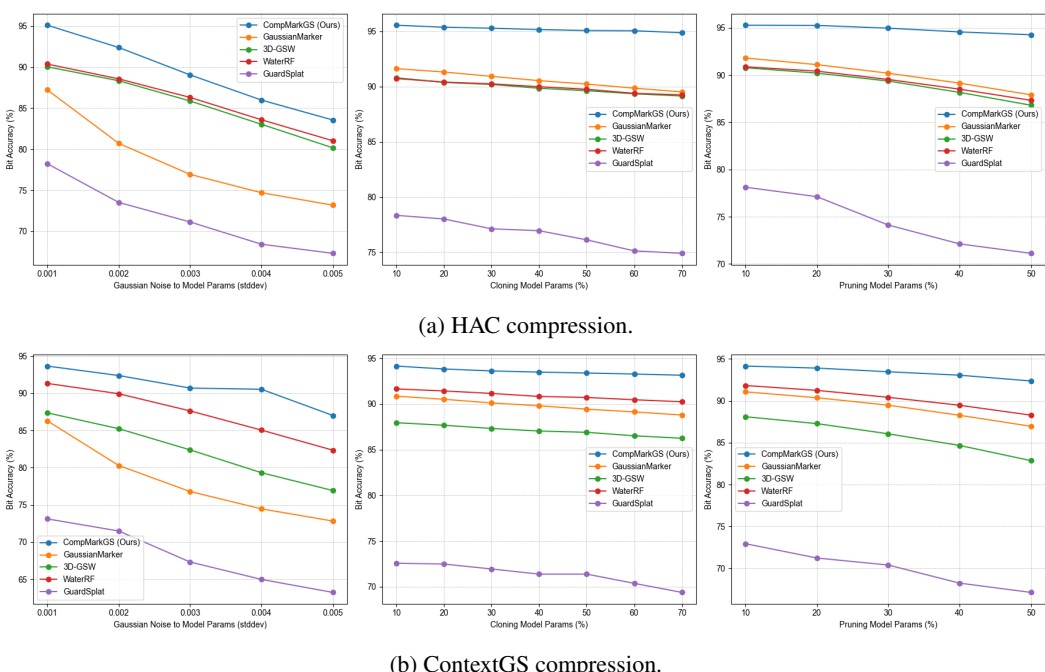

(a) HAC compression.

(b) ContextGS compression.

Figure 9: Comparison of baselines and our method under Gaussian noise, cloning, and pruning. Results are averaged over Blender, LLFF, and Mip-NeRF 360 with 48-bit messages.

### E.7 PERFORMANCE ACROSS COMPRESSION LEVELS

Compression plays a crucial role in determining the trade-off between model size and performance in terms of bit accuracy and rendering quality. Fig.10 shows the relationship among model size, bit accuracy, and rendering quality under two different compression schemes: HAC (Chen et al., 2024a) and ContextGS (Wang et al., 2024a). Fig.10 indicates that a lower compression level leads to better bit accuracy and rendering quality. Our method consistently outperforms other methods, achieving the highest bit accuracy and rendering quality across varying model sizes. Notably, for HAC compression, we achieve bit accuracy of 90% with model size less than 10 MB, while other methods only reach 60% under the same model size. These results highlight the effectiveness and robustness of CompMarkGS.

Table 13: Robustness to model distortions before (left) and after (right) compression.

| Methods | Bit Accuracy(%) ↑ | | | |
| --- | --- | --- | --- | --- |
| | No Distortion | Gaussian Noise ($\sigma = 0.005$) | Clone (50%) | Prune (20%) |
| HAC + WateRF | 91.02 / 54.40 | 81.03 / 53.38 | 89.78 / 54.04 | 87.33 / 53.70 |
| HAC + GaussianMarker | 92.00 / 58.34 | 73.18 / 58.35 | 90.23 / 58.29 | 87.90 / 87.89 |
| HAC + 3D-GSW | 90.96 / 53.48 | 80.17 / 53.11 | 89.62 / 53.17 | 86.80 / 53.18 |
| HAC + GuardSplat | 79.77 / 52.72 | 67.32 / 51.21 | 76.12 / 56.28 | 71.25 / 59.13 |
| HAC + CompMarkGS | **95.95 / 95.92** | **82.13 / 77.13** | **95.33 / 95.09** | **94.67 / 94.25** |
| ContextGS + WateRF | 92.01 / 90.36 | 82.36 / 79.91 | 90.70 / 88.95 | 88.29 / 86.20 |
| ContextGS + GaussianMarker | 91.24 / 87.95 | 72.84 / 71.78 | 89.43 / 86.05 | 86.97 / 83.56 |
| ContextGS + 3D-GSW | 88.28 / 79.75 | 76.94 / 68.55 | 86.89 / 77.99 | 82.87 / 72.70 |
| ContextGS + GuardSplat | 73.20 / 67.16 | 63.24 / 59.84 | 71.39 / 70.11 | 67.12 / 65.19 |
| ContextGS + CompMarkGS | **94.36 / 94.03** | **87.03 / 83.08** | **93.37 / 93.23** | **92.38 / 92.57** |

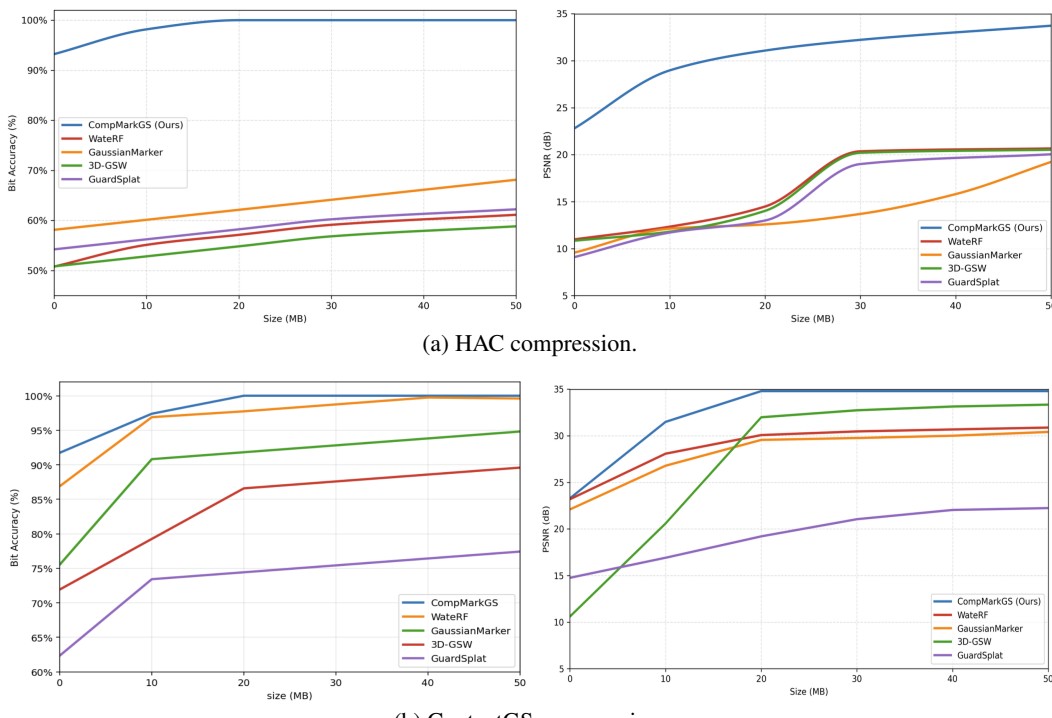

(a) HAC compression.

(b) ContextGS compression.

Figure 10: Performance of bit accuracy and rendering quality under different compression levels. A larger model size corresponds to a lower compression level. The blue line represents the results of our method. Results represent the average score across Blender, LLFF, and Mip-NeRF 360 datasets using 48-bit messages.

## F  COMPUTING RESOURCES

### F.1  OVERHEAD AND EFFICIENCY OF FREQUENCY-AWARE ANCHOR GROWING

To quantify the practical overhead introduced by FAG, we compare CompMarkGS with and without FAG on the Mip-NeRF 360 (Barron et al., 2022) dataset in terms of model size, number of anchors, training time, and rendering speed. As shown in Tab.14, enabling FAG increases the model size and the number of anchors, while the training time grows only slightly. The rendering speed decreases, but it remains well above the 30 FPS real-time threshold. Overall, FAG improves rendering quality and watermark robustness at a modest computational cost, preserving the real-time rendering performance of anchor-based 3DGS.

Table 14: Overhead comparison of CompMarkGS with and without FAG on Mip-NeRF 360.

| Method | Size↓ (MB) | Number of anchors (K) | FPS↑ | Training time↓ (Min.) |
|---|---|---|---|---|
| CompMarkGS (w/o FAG) | 319.79 | 766.4 | 62.27 | 35.26 |
| CompMarkGS (w/ FAG) | 377.02 | 906.6 | 57.34 | 36.36 |

## F.2 REAL-TIME RENDERING PERFORMANCE

We evaluate the rendering performance of our method and baselines (Huang et al., 2025; Jang et al., 2024; 2025) by measuring FPS. As shown in Tab.15, our method exceeds the real-time rendering threshold of 30 FPS across the Blender (Mildenhall et al., 2021), LLFF (Mildenhall et al., 2019), and Mip-NeRF 360 (Barron et al., 2022) datasets. These results demonstrate the practical applicability and rendering efficiency of our proposed CompMarkGS.

Table 15: FPS results on the Blender, LLFF, and Mip-NeRF 360 datasets. Results are averaged over HAC and ContextGS after compression under the 48-bit setting.

| Methods | FPS ↑ | | |
| | Blender | LLFF | Mip-NeRF 360 |
|---|---|---|---|
| WateRF | 199.30 | 19.31 | 41.98 |
| GaussianMarker | 166.45 | 16.24 | 28.54 |
| 3D-GSW | 220.60 | 22.67 | 50.64 |
| GuardSplat | 207.56 | 21.93 | 41.28 |
| CompMarkGS | **230.39** | **35.30** | **67.58** |

## G ADDITIONAL QUALITATIVE RESULTS

Fig.11 visualizes the intermediate state during the frequency-aware anchor growing (FAG) training phase. We compare the high-frequency SSIM-based error map, which highlights candidate regions, with the spatial distribution of new anchors produced by FAG. New anchors do not densely cover the entire error map, but instead appear selectively in areas where high-frequency errors are large enough to exceed the gradient threshold $\tau_{high}^g$. This indicates that FAG does not blindly increase the number of anchors, but focuses anchor growth on structural details that still require refinement.

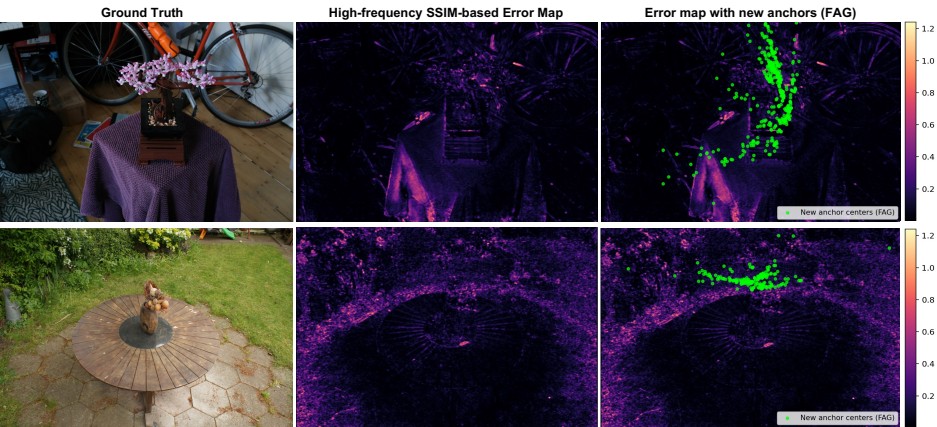

Figure 11: Visualization of an intermediate state of frequency-aware anchor growing (FAG) during training. For the bonsai (top) and garden (bottom) scenes in the Mip-NeRF 360 dataset, we visualize the ground-truth image, the high-frequency SSIM-based error map, and the same error map with new anchors (green). New anchors are mainly added along sparse high-frequency structures with large SSIM error, rather than uniformly over the image, demonstrating that FAG selectively refines regions that still exhibit high rendering error.

Fig.12 through Fig.20 visualize all results rendered after compression using our method with HAC (Chen et al., 2024a) and ContextGS (Wang et al., 2024a), along with the difference (×5) between the original images and the watermarked images.

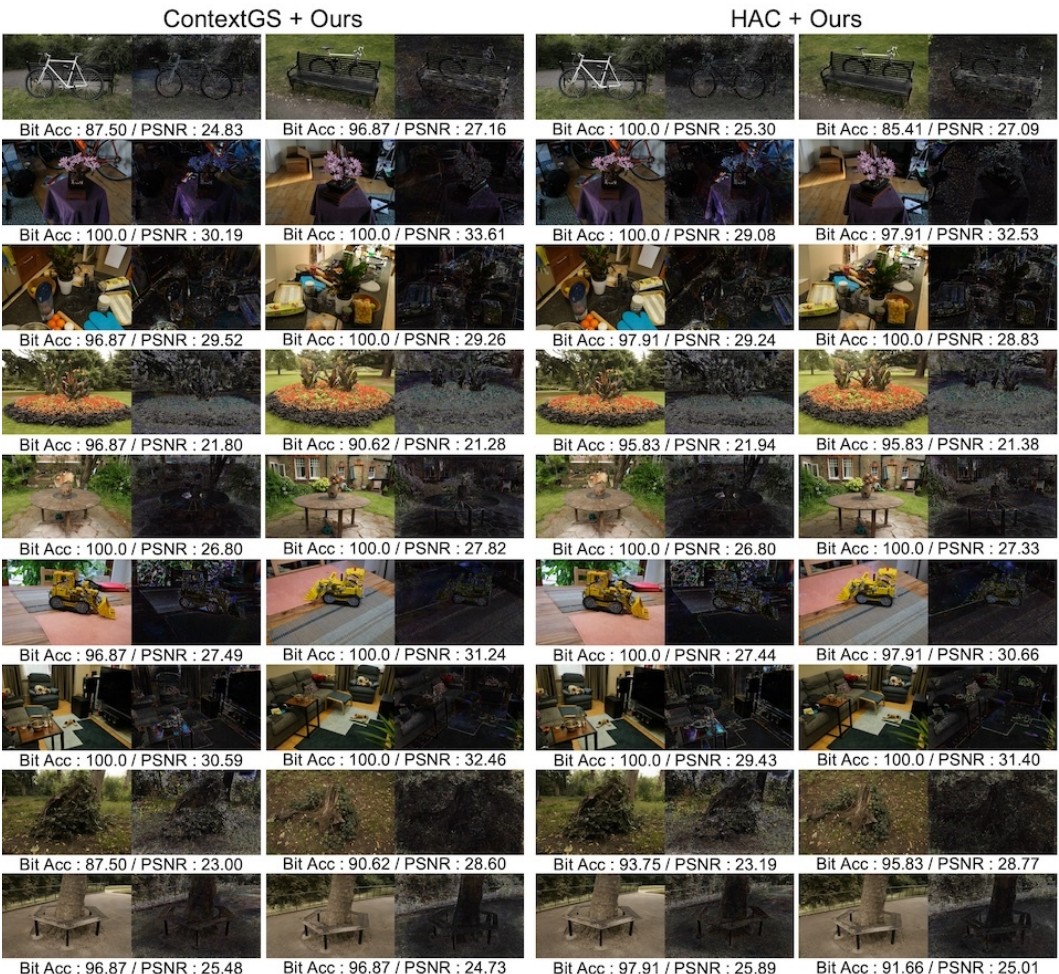

Figure 12: Rendering quality of various rendering outputs generated by our method on the Mip-NeRF 360 dataset. We show the differences (× 5). The closer it is to white, the greater the discrepancy between the ground truth and the rendered image. The results were obtained using 32-bit messages after compression.

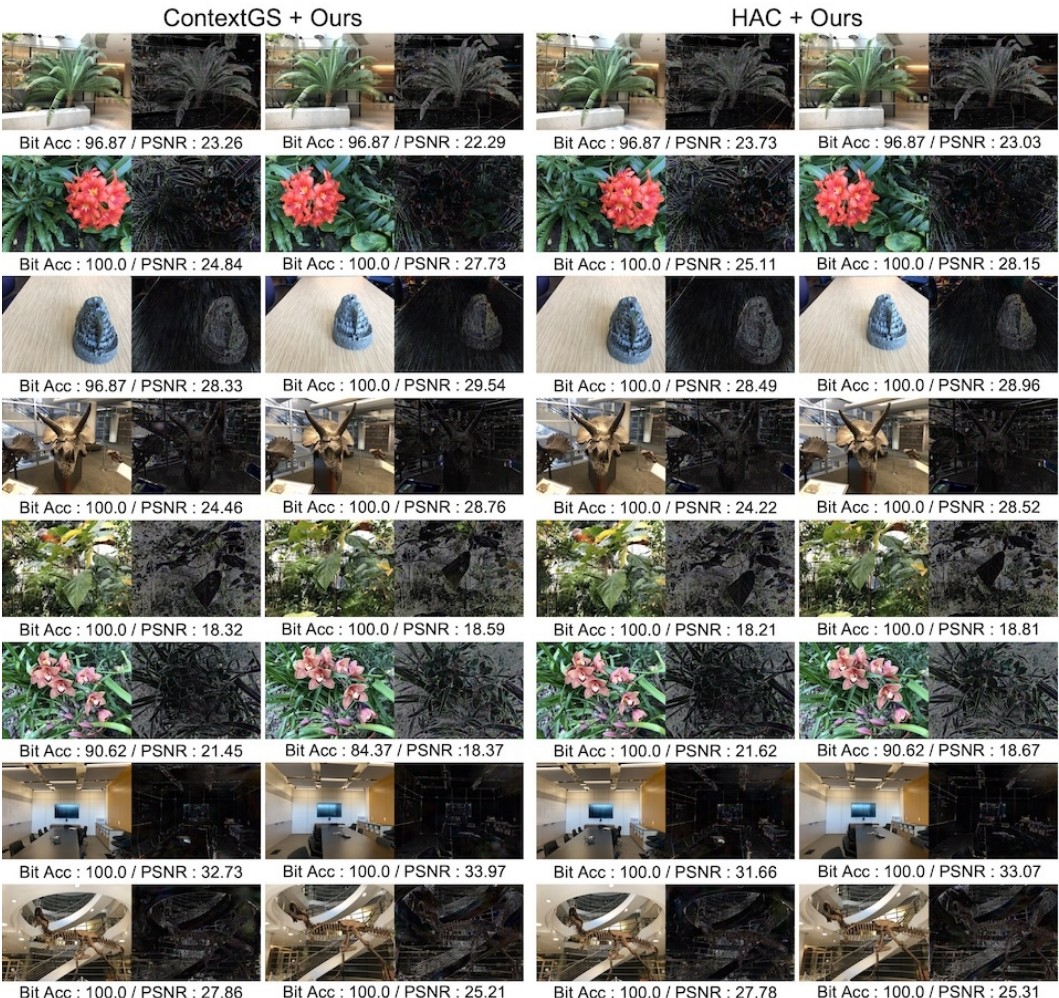

Figure 13: Rendering quality of various rendering outputs generated by our method on the LLFF dataset. We show the differences (× 5). The closer it is to white, the greater the discrepancy between the ground truth and the rendered image. The results were obtained using 32-bit messages after compression.

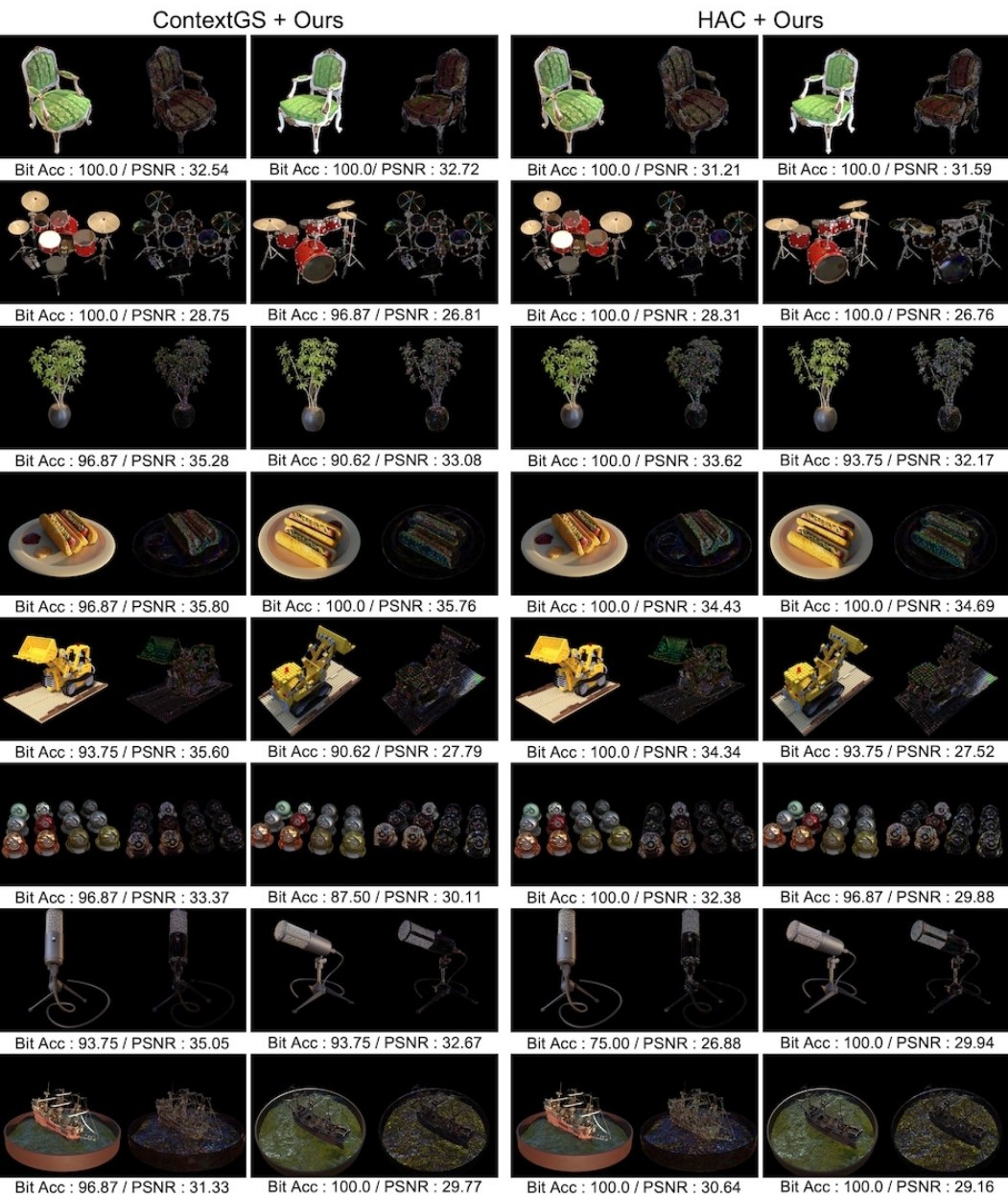

Figure 14: Rendering quality of various rendering outputs generated by our method on the Blender dataset. We show the differences (×5). The closer it is to white, the greater the discrepancy between the ground truth and the rendered image. The results were obtained using 32-bit messages after compression.

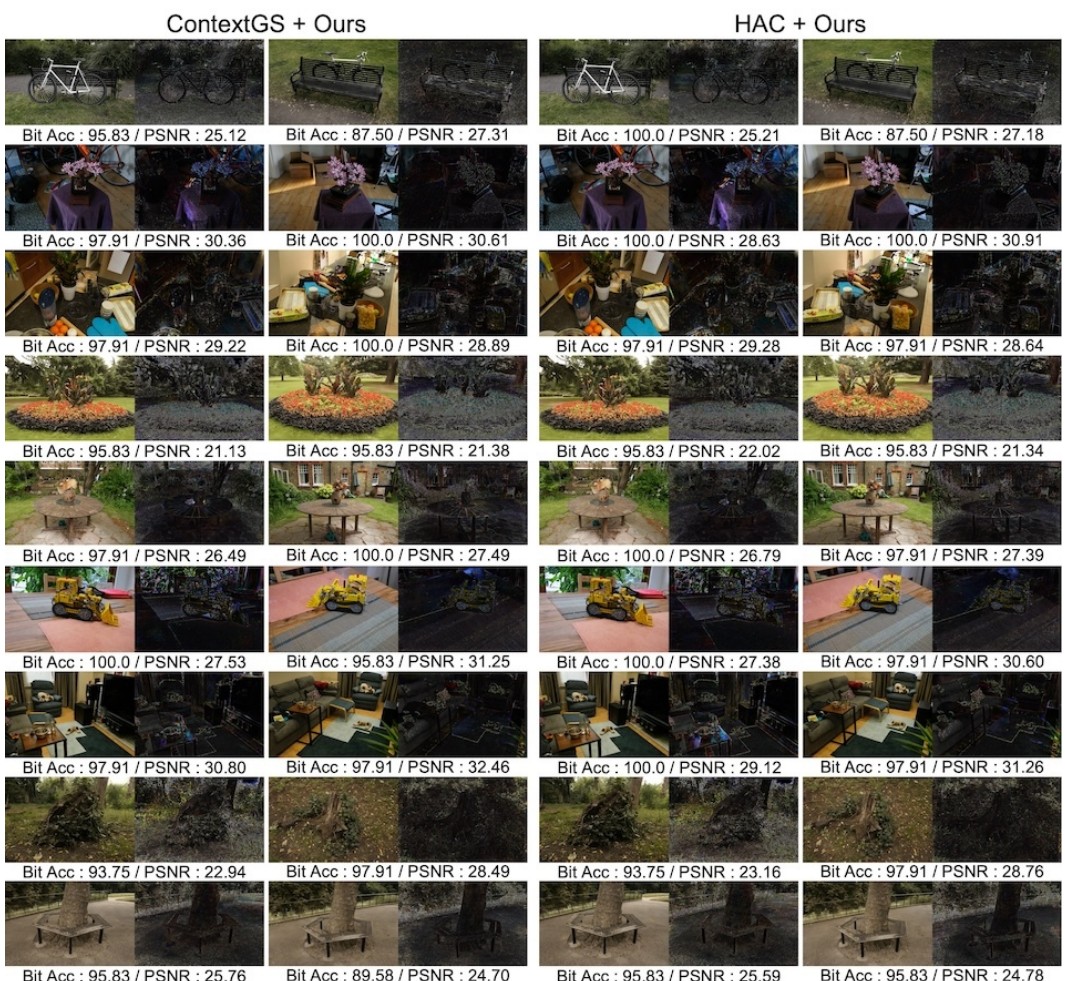

Figure 15: Rendering quality of various rendering outputs generated by our method on the Mip-NeRF 360 dataset. We show the differences (× 5). The closer it is to white, the greater the discrepancy between the ground truth and the rendered image. The results were obtained using 48-bit messages after compression.

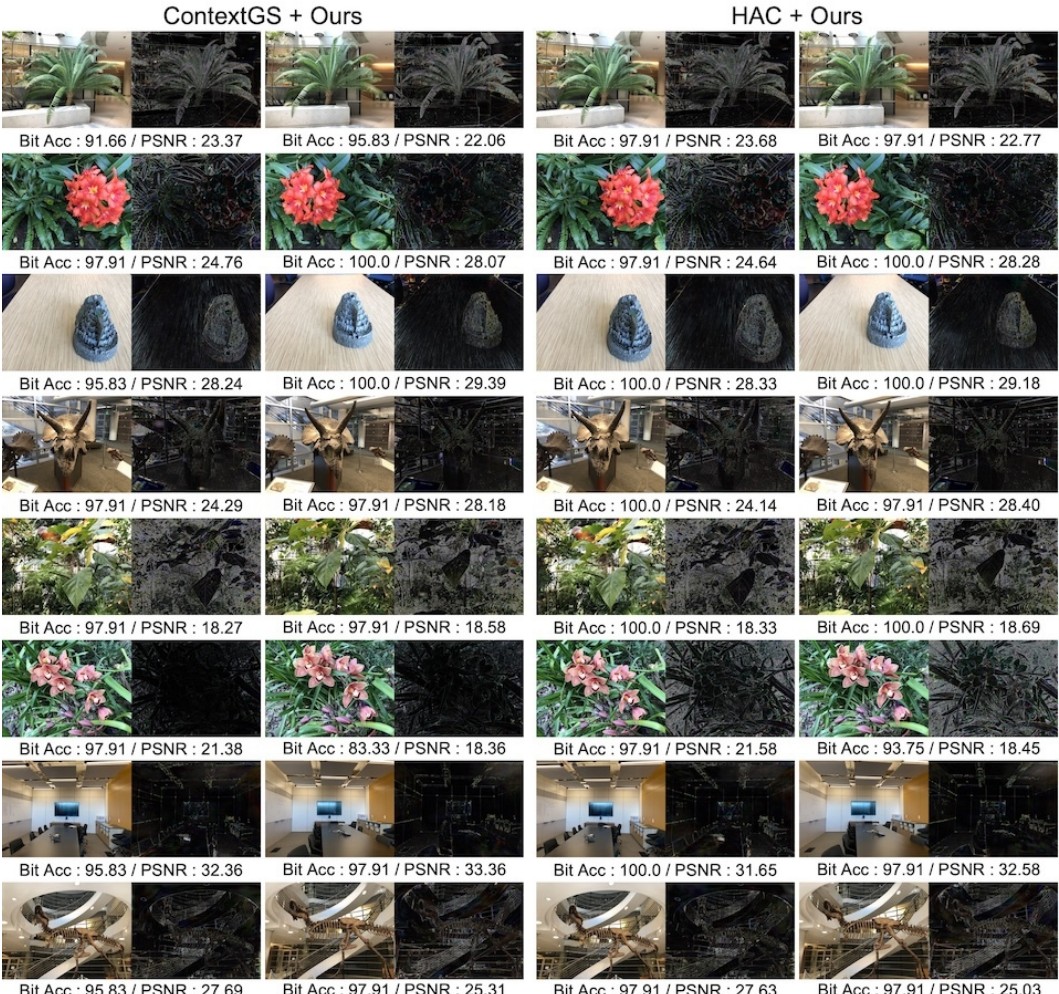

Figure 16: Rendering quality of various rendering outputs generated by our method on the LLFF dataset. We show the differences (× 5). The closer it is to white, the greater the discrepancy between the ground truth and the rendered image. The results were obtained using 48-bit messages after compression.

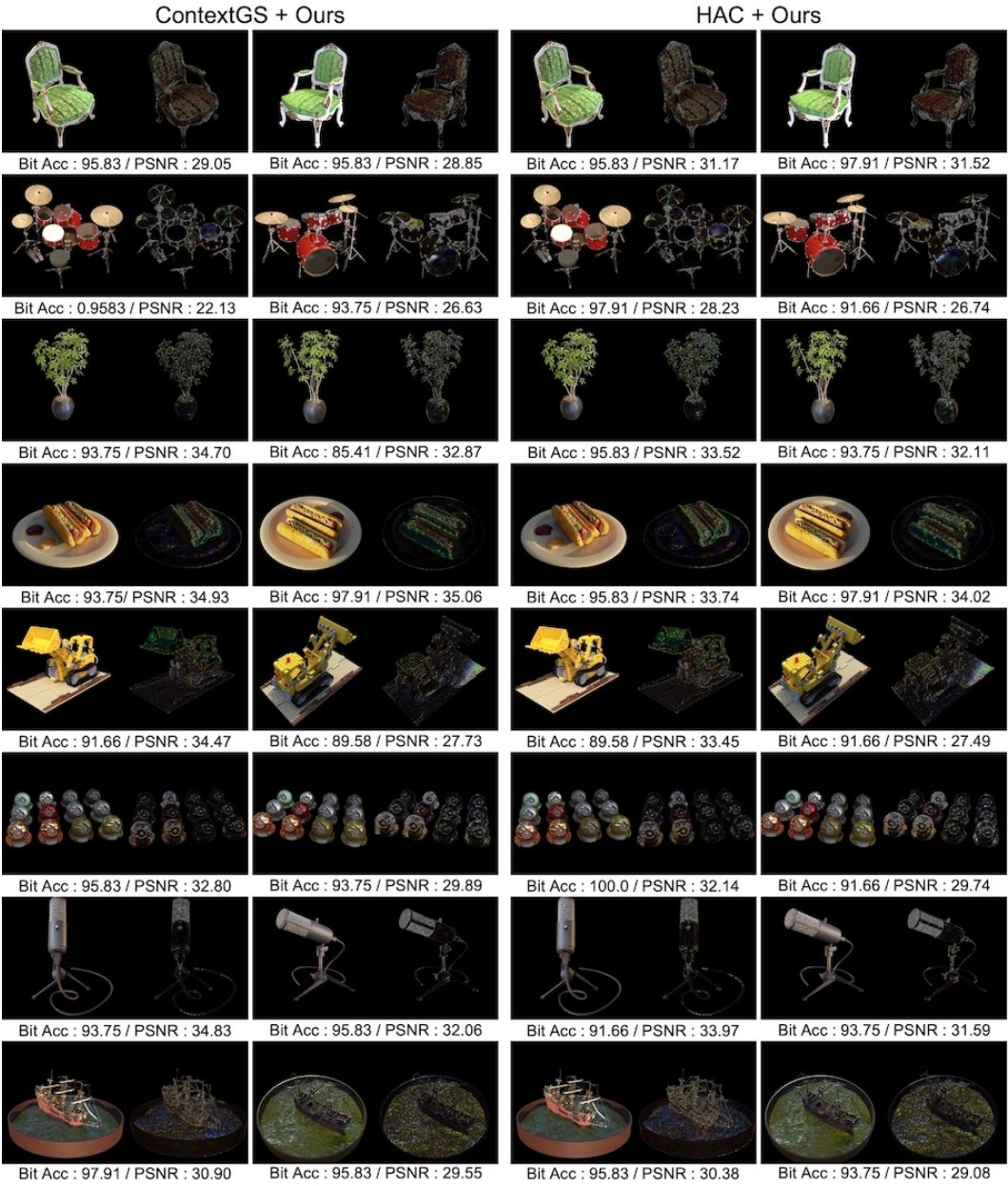

Figure 17: Rendering quality of various rendering outputs generated by our method on the Blender dataset. We show the differences (× 5). The closer it is to white, the greater the discrepancy between the ground truth and the rendered image. The results were obtained using 48-bit messages after compression.

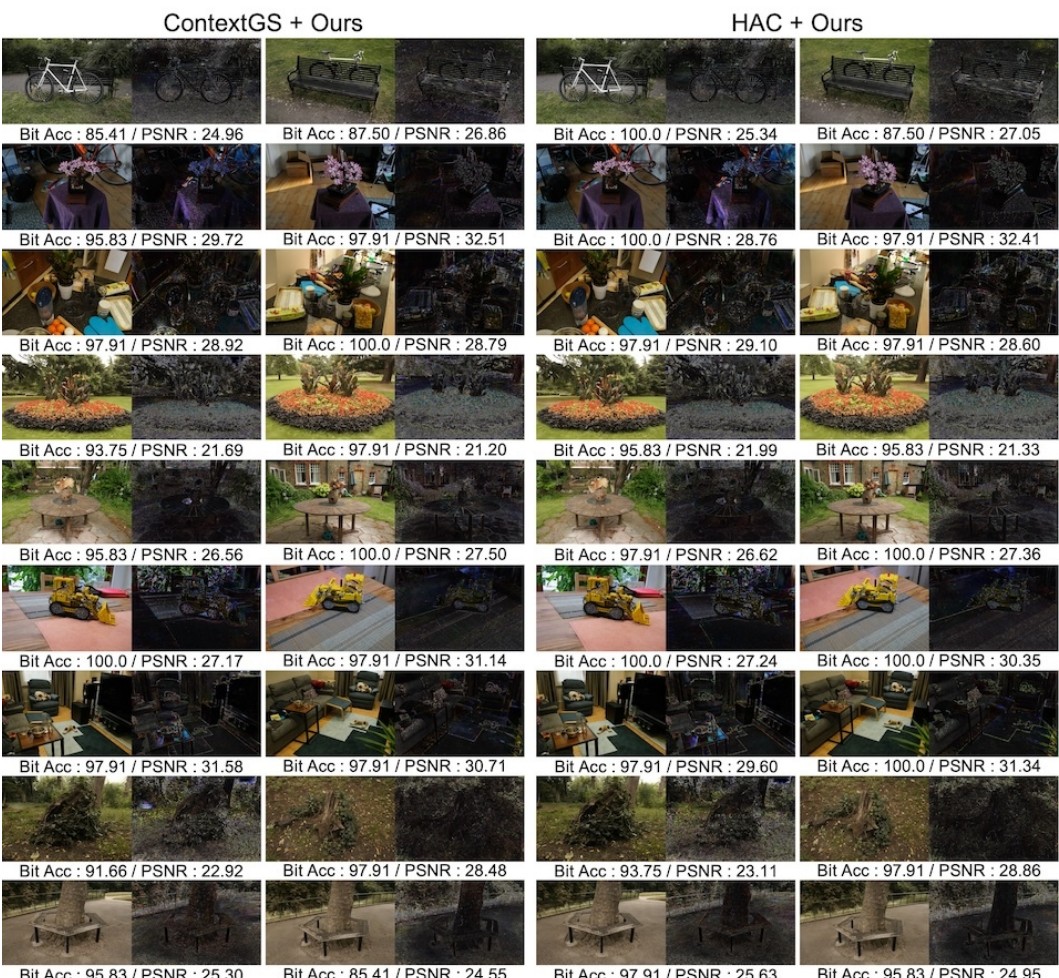

Figure 18: Rendering quality of various rendering outputs generated by our method on the Mip-NeRF 360 dataset. We show the differences (× 5). The closer it is to white, the greater the discrepancy between the ground truth and the rendered image. The results were obtained using 64-bit messages after compression.

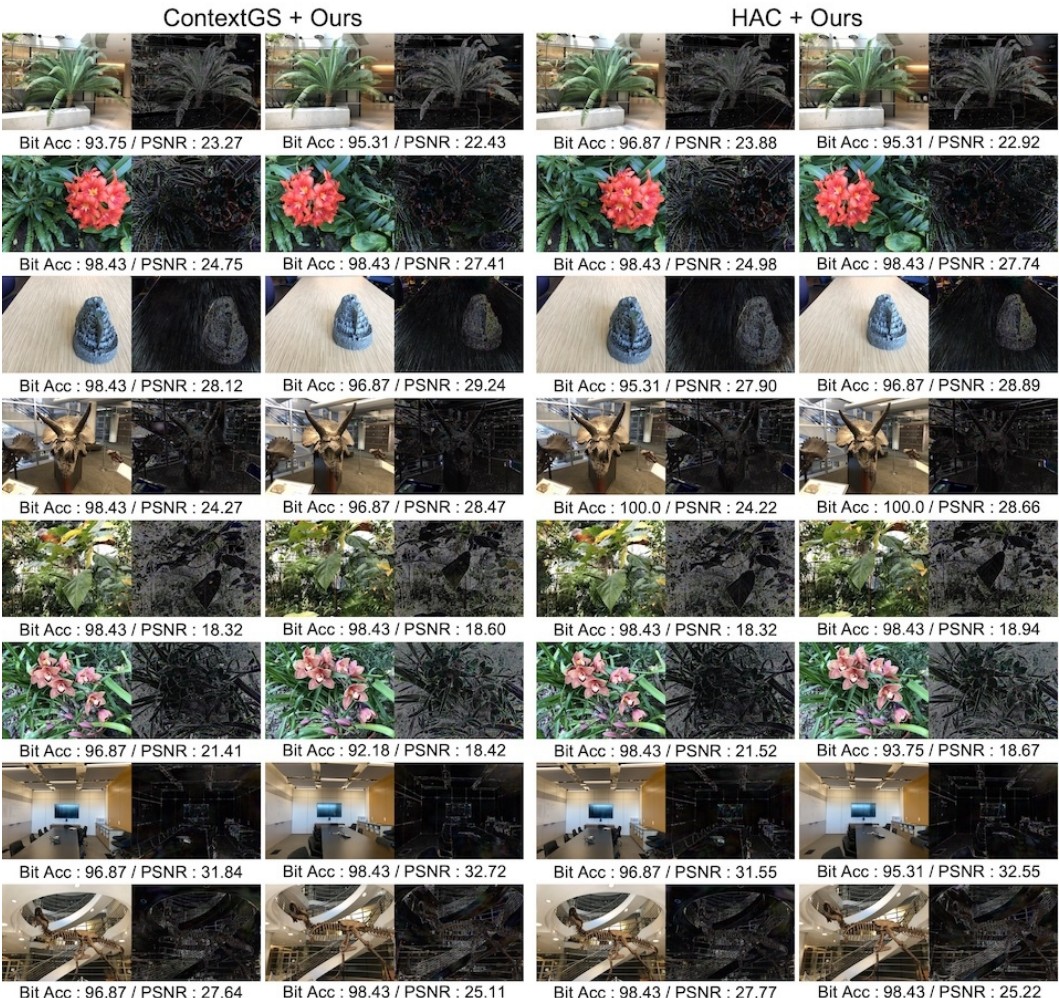

Figure 19: Rendering quality of various rendering outputs generated by our method on the LLFF dataset. We show the differences (× 5). The closer it is to white, the greater the discrepancy between the ground truth and the rendered image. The results were obtained using 64-bit messages after compression.

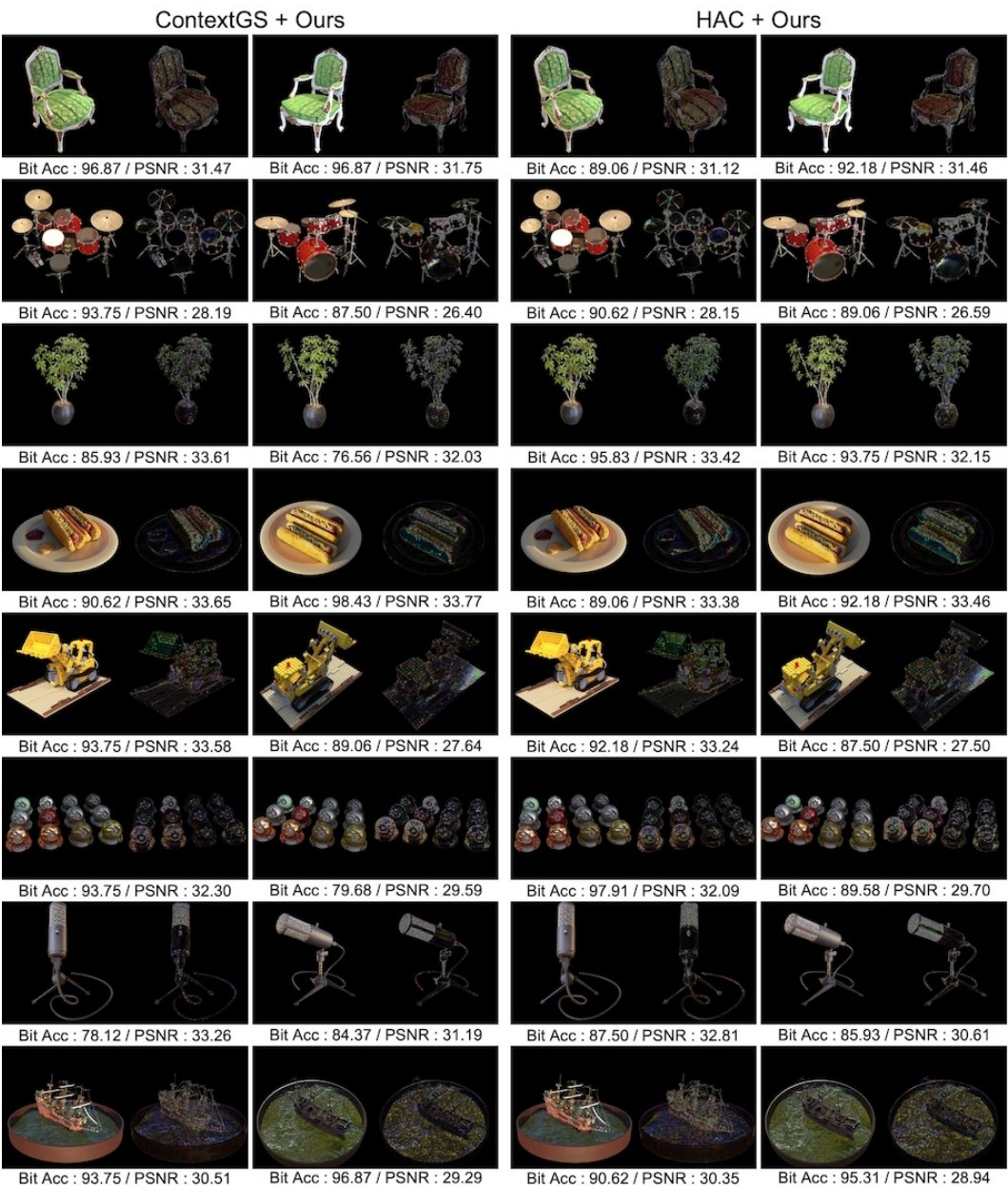

Figure 20: Rendering quality of various rendering outputs generated by our method on the Blender dataset. We show the differences (× 5). The closer it is to white, the greater the discrepancy between the ground truth and the rendered image. The results were obtained using 64-bit messages after compression.

