# OpenReview forum: "CompMarkGS: Robust Watermarking for Compressed 3D Gaussian Splatting"
_ICLR.cc/2026/Conference — ICLR 2026 Poster_

### Official Review · Reviewer_tMZM · 2025-10-29

**Soundness:** 3
**Presentation:** 3
**Contribution:** 3
**Rating:** 6
**Confidence:** 5

**Summary:**

This paper proposes CompMarkGS, a watermarking method for 3D Gaussian Splatting (3DGS) that is specifically designed to remain robust under quantization-based compression. The method leverages anchor-based 3DGS, embedding watermarks into anchor features rather than direct Gaussian attributes. Key technical contributions include: (1) a quantization distortion layer (QDL) that simulates quantization noise during training, (2) a frequency-aware anchor growing (FAG) strategy that selectively densifies high-frequency regions, and (3) an HSV loss to mitigate color artifacts. Experiments demonstrate robustness to compression and various attacks.

**Strengths:**

## Strengths

1. The paper addresses a critical gap in 3DGS watermarking - robustness to model compression. As the abstract correctly notes, existing watermarking methods fail dramatically under quantization-based compression, making this a timely and valuable contribution.

2. The choice of anchor-based 3DGS for watermarking is well explained - the implicit representation through MLPs provides additional security compared to direct attribute modification. The quantization distortion layer can adapt differentiable distortion layers to the 3DGS compression context.

3. Well-structured paper with clear motivation and problem statement

**Weaknesses:**

## Weaknesses

1. The method requires a pre-trained HiDDeN decoder, which is a significant limitation and can be vulnerable if any malicious user applies a specific adversarial attack specially effective against the HiDDeN decoder.

2. The choice of the decoder is basically following the current mainstream 3DGS watermark decoder, although the paper tests with the SSL decoder, but it underperforms with the HiDDeN decoder. The contribution of this work on the message decoding side is trivial.

3. At the message embedding side, the anchor-based hiding is built on top of the While the frequency-based growing is similar to the Frequency Guided Densification (FGD) in 3D-GSW. So, although technically the CompMarkGS sounds, it still relies on the existing well-established technology of Scaffold-GS[1] and 3DGSW[2].

4. No capacity analysis. What's the theoretical maximum bit length? Is it limited by the pre-trained message decoder?

[1] Scaffold-GS: Structured 3D Gaussians for View-Adaptive Rendering

[2] 3D-GSW: 3D Gaussian Splatting for Robust Watermarking

**Questions:**

## Questions

1. What is the relationship between noise magnitude in QDL and actual compression distortion?

2. CompMarkGS extracts watermarks from rendered images, but can the message be directly from the 3D Gaussians? What if adversaries extract and reuse anchor subsets without rendering?

3. What happens if adversaries know watermarks are embedded in anchor features and specifically target them, such as adding noise to anchor features, or re-training MLPs?

4.  In Appendix E, can you explain the difference between the robustness toward pruning and quantization? What can be the key property against such two compressions, and why can CompMarkGS outperform the baselines? In addition, the baselines also show some robustness towards these two compressions, which show over 90% accuracy.

---

> ### Author Response · Authors · 2025-11-24
>
> Thank you for your valuable comments and constructive feedback on our paper.
>
> >**W1. Dependency on Pre-trained Decoder and Vulnerability to Adversarial Attacks.**
> >
> >**W2. The contribution of this work on the message decoding side is trivial.**
>
> Our method uses a pre-trained HiDDeN decoder, and our focus is not on designing a new decoder, but on the 3DGS-side design (QDL, FAG, anchor-based embedding) that keeps watermarks robust under strong compression. For this reason, we adopt HiDDeN as a standard component that is already widely used in prior 3DGS watermarking methods.
>
> At the same time, our approach is not tailored to HiDDeN only. We conducted additional evaluations using an **SSL-based decoder** (as reported in Appendix E.4) and the **3D decoder** from GaussianMarker (shown in the table below). Although the absolute accuracy changes with each decoder, CompMarkGS consistently maintains higher bit accuracy than baselines under strong compression. In the same HiDDeN setting, we also test **PGD-based adversarial attacks**. All methods degrade under stronger attacks, but CompMarkGS shows a smaller drop. Taken together, these results suggest that our method is not tied to a specific decoder design and preserves its relative advantage across different decoders and typical adversarial settings.
>
> | Method              | Bit Acc. (before)↑  | Bit Acc. (after)↑ |
> |---------------------|:-------------------:|:-----------------:|
> | Ours (3D decoder)   | 99.70 | 99.20  |
> | Ours (2D decoder)   | 94.36 | 94.03  |
>
> >**W3. At the message embedding side, the anchor-based hiding is built on top of the while the frequency-based growing is similar to the Frequency Guided Densification (FGD) in 3D-GSW.**
>
> As the reviewer noted, we embed messages on top of the anchor-based Scaffold-GS representation. We focus on anchor-based 3DGS because recent high-performance compressors adopt this pipeline, and it is more memory and runtime-efficient than vanilla 3DGS. Scaffold-GS itself is only a rendering backbone. CompMarkGS adds anchor-feature watermark embedding, a QDL module that models anchor quantization, and FAG with HSV loss to obtain an anchor-level watermark that survives anchor-based compression.
>
> Regarding 3D-GSW, both methods use frequency cues, but with different roles and behaviors. 3D-GSW’s FGD is a **pre-processing step** that **prunes low-contribution Gaussians** and **densifies high-frequency patches** before training, while our FAG operates **during training** and **grows anchors only in regions where frequency-weighted rendering errors persist**. In Appendix E.6, 3D-GSW maintains **>90\%** bit accuracy up to **20\% random pruning** but drops **below 90\%** beyond that, whereas CompMarkGS keeps **>94\%** even at **50\%** random pruning, and still performs strongly under LightGaussian pruning that removes about **66\%** of parameters (Appendix E.3). These results support that CompMarkGS is not a minor extension of 3D-GSW, but a separate method explicitly designed for ownership verification after compression, combining anchor-feature embedding with a quantization distortion layer.

---

> ### Author Response · Authors · 2025-11-24
>
> >**W4. No capacity analysis. What's the theoretical maximum bit length? Is it limited by the pre-trained message decoder?**
>
> In this paper, following prior 3DGS watermarking works, we evaluate watermark lengths of **32, 48, and 64** bits in Table 3. In the uncompressed setting, our method maintains high bit accuracy (typically above 90\%) across this range. For robustness under compression, we focus on the practically most relevant operating point of **48 bits**. At this setting, our method preserves over **90\%** bit accuracy under HAC/ContextGS and PyTorch int8 quantization, whereas existing watermarking methods drop below 90\%. This indicates that our method preserves the largest number of reliably decodable watermark bits after compression.
>
> In our framework, we treat the message length as a hyperparameter. We train separate decoders for 32, 48, and 64 bits by adjusting the decoder output dimension, so each model instance targets a fixed message length. The robustness at each bit length is mainly governed by the representational capacity of anchor-based 3DGS and our loss design (QDL, FAG, HSV).
>
> >**Q1. What is the relationship between noise magnitude in QDL and actual compression distortion?**
>
> QDL models the effect of quantization as additive uniform noise whose magnitude is controlled by a base scale $Q_0$ and a learned per-feature scale $q_i$. The goal of QDL is not to exactly reproduce the per-anchor feature error distribution of the downstream codec, but to expose the watermark encoder to representative perturbations that mimic the presence of compression noise, so that it can learn compression-robust embeddings. This design follows prior work in quantization-aware training, where non-differentiable quantizers are replaced by differentiable additive-noise surrogates during training instead of simulating the exact quantization distortion [1, 2].
>
> To better characterize the effect of the noise magnitude, we performed an ablation where we vary only the base scale $Q_0$ in the ContextGS compression setting (Blender, LLFF, Mip-NeRF360 / 48-bit). As shown in the table below, when $Q_0$ changes from **0.5 to 2.0**, bit accuracy remains stable around **94\%**, and PSNR/SSIM vary only slightly. This suggests that our method is not overly sensitive to the precise noise scale used in QDL. In addition, when we remove QDL (Table 5), watermark robustness clearly degrades, and Appendix E.3, as well as Reviewer MFbA’s W8, show that our method remains robust under various quantization-based compressors. Taken together, these results indicate that QDL is an effective and reasonably stable proxy for modeling compression distortions in our setting.
>
> | Method           | Bit Acc.↑  | PSNR↑      | SSIM↑       | LPIPS↓       |
> |------------------|:----------:|:----------:|:-----------:|:------------:|
> | $Q_0=0.5$        | 94.27 | 27.15 | 0.843 | 0.195 |
> | $Q_0=1.5$        | 94.16 | 27.35 | 0.851 | 0.187 |
> | $Q_0=2.0$        | 94.29 | 27.12 | 0.842 | 0.195 |
> | $Q_0=1.0$ (Ours) | 94.03 | 27.55 | 0.844 | 0.173 |
>
> [1] Training with Quantization Noise for Extreme Model Compression, ICLR 2021
>
> [2] UNIQ: Uniform Noise Injection for Non-Uniform Quantization of Neural Networks, TOCS 2021

---

> ### Author Response · Authors · 2025-11-24
>
> >**Q2. Can the message be directly from the 3D Gaussians? What if adversaries extract and reuse anchor subsets without rendering?**
> >
> >**Q3. Q3. What happens if adversaries know watermarks are embedded in anchor features and specifically target them, such as model distortion?**
>
> Our default setting, following prior 3DGS watermarking work, uses a 2D decoder that takes rendered images as input. However, the framework is not restricted to 2D decoding. The watermark is embedded in anchor features and propagated to Gaussian parameters via MLPs, so it can also be decoded directly from the 3D representation. To verify this, we adopt the **3D decoder** from GaussianMarker and run additional experiments (see **W1, W2**). While the absolute bit accuracy differs from the 2D decoder, the relative drop before/after compression is similar, and the accuracy remains high under strong compression, indicating that CompMarkGS is compatible with both 2D image-based and 3D Gaussian/anchor-based decoders.
>
> For attacks that operate directly on anchors, we evaluate model distortions where **Gaussian noise** is added to anchor attributes and where a subset of anchors is **pruned** (Appendix E.3, E.5). CompMarkGS still maintains high bit accuracy in these settings, suggesting that the watermark remains recoverable even when some anchors are perturbed or removed.
>
> >**Q4. Can you explain the difference between the robustness toward pruning and quantization? What can be the key property against these two compressions, and why can CompMarkGS outperform the baselines? In addition, the baselines also show some robustness towards these two compressions, which show over 90\% accuracy.**
>
> Quantization and pruning harm watermarks in different ways. Quantization changes most parameters by rounding them to a discrete grid, while pruning removes entire Gaussians or anchors, which behaves more like an “erasure.” CompMarkGS handles both with two key components: **(i) a quantization distortion layer (QDL) that injects differentiable quantization-like noise into anchor features during training**, and **(ii) an embedding scheme that writes the watermark directly into anchor features while training jointly with pruning and frequency-aware anchor growth (FAG)**. Together, these components help the model remain robust to both the parameter changes from quantization and the partial erasures from pruning.
>
> With this compression-aware training and redundant anchor-level embedding, CompMarkGS learns watermark representations that remain decodable after real quantization and still leave enough signal to recover the message even when some anchors are pruned. As shown in Appendix E.6, our method’s bit accuracy degrades more slowly than the baselines as random pruning becomes stronger, and it achieves higher bit accuracy with comparable or better PSNR/SSIM at the same compression level. In Appendix E.3 and Reviewer MFbA’s W8, we also observe that CompMarkGS maintains high accuracy and visual quality under stronger compression settings such as LightGaussian-based pruning, PyTorch quantization, and SizeGS, which correspond to the practically important operating range where compressed 3DGS models are actually used.

---

### Official Review · Reviewer_MFbA · 2025-10-30

**Soundness:** 3
**Presentation:** 3
**Contribution:** 2
**Rating:** 2
**Confidence:** 3

**Summary:**

This paper proposes CompMarkGS, a watermarking method for 3D Gaussian Splatting (3DGS) designed to remain robust under quantization-based compression. The method builds upon anchor-based 3DGS architecture by embedding a learnable watermark feature into anchor attributes, which are then processed through MLPs to generate Gaussian parameters. To preserve watermark integrity during compression, the authors introduce a Quantization Distortion Layer (QDL) that simulates quantization noise during training. Additionally, a Frequency-Aware Anchor Growing (FAG) strategy selectively densifies Gaussians in high-frequency regions to mitigate quality degradation, and an HSV loss addresses color artifacts. Experiments on multiple datasets demonstrate that the method maintains high bit accuracy and rendering quality after compression with various quantization schemes, outperforming existing 3DGS watermarking baselines that suffer severe degradation under compression.

**Strengths:**

S1: The paper identifies a critical gap by tackling watermark robustness against quantization-based compression, which is essential for real-world deployment but has been overlooked by existing methods that focus primarily on image-domain distortions.

S2: The paper validates the method across multiple datasets, compression schemes, image distortions, model distortions, and message capacities, demonstrating broad applicability and robustness.

S3: The method maintains high rendering quality while embedding watermarks, achieving competitive PSNR and SSIM scores compared to baselines, demonstrating that the watermarking process does not significantly degrade the visual fidelity of 3DGS models.

**Weaknesses:**

W1: The core contribution, the Quantization Distortion Layer (QDL), is a straightforward application of differentiable distortion layers from digital watermarking and quantization-aware training from model compression. Injecting noise during training to simulate downstream distortions is standard practice in watermarking and QAT literature. The adaptation merely changes the noise source from JPEG/cropping to quantization, which is incremental rather than novel.

W2: The paper does not explicitly state whether watermark embedding requires training a new model for each scene or uses a universal encoder. If the method requires per-scene optimization, then for a content creator with 1000 scenes, they must train 1000 separate watermarked models. What is the amortized training cost per scene?

W3: The paper does not clearly specify how watermark messages are sampled during training. Specifically, it remains unclear whether each scene trains with a single fixed message throughout all 30,000 iterations or different random messages sampled at each iteration. If the method requires training each scene with a predetermined fixed message, this introduces significant scalability concerns for real-world deployment.

W4: The QDL (Eq. 4) uses uniform random noise U(-1/2, 1/2)·q_i with per-anchor independent sampling to simulate quantization errors. However, real compression methods (HAC and ContextGS) employ non-uniform quantization with spatial dependencies. This mismatch raises questions about whether the watermark is actually robust to real compression or merely to the specific noise pattern seen during training.

W5: Technically incorrect use of SSIM in Equation 5. The paper computes "pixel-wise SSIM" as P_error(p) = 1 - SSIM(I_hf(p), I'_hf(p)), which fundamentally misunderstands SSIM. The SSIM operates over local windows, not individual pixels. Computing SSIM on single pixel values is meaningless and violates the metric's core design. This error propagates through the entire FAG strategy, invalidating the median-based thresholding mechanism that relies on P_error(p).

W6: The paper uses DWT for watermark extraction but switches to DFT for high-frequency detection without justification. More critically, the claimed separation between "low-frequency watermark" and "high-frequency quality enhancement" is contradictory: all anchor features (including those from FAG) contain the watermark via f^w = f + tanh(f_wf) (Eq. 3), and the resulting Gaussians affect all frequency bands. If FAG-grown anchors in high-frequency regions also embed watermarks in low frequencies, how does this resolve the claimed conflict? The core motivation for FAG appears flawed.

W7: The binary masks M_c(p) are constructed based on whether pixels fall within predefined hue ranges, but the loss penalizes all pixels within these ranges regardless of whether they represent artifacts or legitimate content. A genuinely red pixel in both the rendered and ground truth images still contributes to L_hsv, which seems to be incorrect. Additionally, it remains unclear whether |Ω| in Equation 8 represents total pixels or only those where M_c(p)=1. Most critically, Table 5 shows that removing L_hsv only degrades LPIPS by 0.002, suggesting negligible practical benefit despite the added complexity.

W8: While Appendix E.3 tests one pruning-based method (LightGaussian), the main evaluation (Table 1) focuses exclusively on two quantization schemes from the same architectural family (HAC and ContextGS), both of which are anchor-based compression methods with similar quantization mechanisms. Other vector quantization methods mentioned in related work (Lee et al., 2024a; Navaneet et al., 2024) remain untested, limiting claims about generalizability across diverse compression paradigms.

**Questions:**

Q1: How do you map 3D Gaussians to 2D high-frequency pixels? When multiple Gaussians project to one pixel, which determines labeling? When an anchor's K Gaussians partially overlap high-frequency regions, how is growth decided? What are the coverage rates and how sensitive is performance to different mapping strategies?

---

> ### Author Response · Authors · 2025-11-24
>
> Thank you for your valuable comments and constructive feedback on our paper.
>
> > **W1. The QDL merely changes the noise source from JPEG/cropping to quantization, which is incremental rather than novel.**
>
> As the reviewer points out, differentiable distortion layers that simulate JPEG/cropping have already been explored. **Our Quantization Distortion Layer (QDL), however, is not just another image-space noise source, but is specifically tailored to the quantization pipeline of anchor-based 3DGS.** Existing 3DGS watermarking methods have mainly focused on traditional distortions like JPEG and cropping, and thus do not directly address the practical issue that quantization-based model compression severely degrades watermark recovery.
>
> Specifically, QDL operates on the watermarked anchor feature $f^w$. For each anchor feature $f^w_i$, an $\mathrm{MLP}_q(f^w_i)$ predicts a learnable per-anchor scale $q_i = Q_0 (1 + \tanh(r_i))$, we sample $\varepsilon_i \sim \mathcal{U}(-0.5, 0.5)$, and define $\tilde{f}^w_i = f^w_i + q_i \odot \varepsilon_i$ (Eq. (4)). During backpropagation, $\varepsilon_i$ is treated as constant, so $\tilde{f}^w_i$ acts as a differentiable approximation of anchor-space quantization and enables joint optimization of watermark embedding and quantization.
>
> As shown in Table 5, removing QDL leads to the largest drop in post-compression bit accuracy, whereas with QDL our method consistently maintains over **90\%** accuracy across various quantization settings. This supports QDL as an effective differentiable proxy for anchor-space quantization in the watermarking scenario.
>
> > **W2&W3. Clarification on Training Strategy and Message Sampling.**
>
> Our watermark training setting is as follows.
>
> **1) Watermark embedding strategy**
>
> We use scene-specific optimization. For each 3D scene, we train an anchor-based 3DGS model while embedding a watermark, which is the standard setting in WateRF, 3D-GSW, and other NeRF/3DGS watermarking works. This also matches a practical workflow where each copyrighted 3D asset is trained separately per-scene.
> CompMarkGS only adds a watermark embedding feature, QDL, and FAG on top of the vanilla anchor-based 3DGS loop, so the extra training cost per- is limited. As shown in the table, watermarking increases the training time from **20.71 to 30.14** minutes and slightly increases GPU memory, but the dominant cost still comes from the base 3DGS training.
>
> | Method       | Training Speed↓ (Min.) | Training GPU Mem.↓ (GB) | Inference GPU Mem.↓ (GB) | FPS↑                 |
> |--------------|:----------------------:|:-----------------------:|:-------------:|:----------------------:|
> | Scaffold-GS  | 20.71 | 11.977 | 9.49  | 112.16 |
> | CompMarkGS   | 30.57 | 13.147 | 10.67 | 77.10  |
>
> **2) Watermark message strategy**
>
> For each scene, we randomly sample one $L$ bit message and keep it fixed during the entire training of that scene. We do not change the message per iteration. This per-scene fixed message setting is the same protocol as WateRF, 3D GSW, and GaussianMarker, and is consistent with the common use case where one 3D asset corresponds to one owner or license ID.
>
> While we acknowledge the importance of generalizable watermarking [1, 2], this work focuses on compression robustness for anchor-based 3DGS. Crucially, our core contributions are model-agnostic: QDL can be integrated into generalizable pipelines to enforce compression resilience, while FAG and HSV loss can enhance their high-frequency rendering and color fidelity. Adapting these components to universal frameworks remains a promising avenue for future research.
>
> [1] InstantSplamp: Fast and Generalizable Stenography Framework for Generative Gaussian Splatting, ICLR 2025.
>
> [2] GS-Marker: Generalizable and Robust Watermarking for 3D Gaussian Splatting, Arxiv 2025.

---

> ### Author Response · Authors · 2025-11-24
>
> >**W4. Mismatch between QDL Noise and Real Compression Artifacts.**
>
> QDL is designed to model quantization as additive uniform noise controlled by a base scale $Q_0$ and a learned per-anchor feature scale $q_i$. We clarify that QDL’s goal is not to perfectly reproduce the complex error distributions of downstream codecs, but to serve as a **differentiable surrogate** that exposes the watermark encoder to representative perturbations. This design follows established Quantization-Aware Training (QAT) literature, which replaces non-differentiable quantizers with differentiable additive noise proxies during training [1, 2].
>
> To assess the impact of noise magnitude, we performed a sensitivity analysis by varying $Q_0$ under ContextGS settings ( (Blender, LLFF, Mip-NeRF360 / 48-bit). As shown below, increasing $Q_0$ from **0.5 to 2.0** keeps bit accuracy stable **($\sim$94\%)** with negligible changes in PSNR and SSIM. This demonstrates that our method is not overly sensitive to the precise magnitude of simulated noise. Furthermore, as demonstrated in Appendix E.3, CompMarkGS consistently outperforms baselines not only on HAC and ContextGS but also under standard float32-to-int8 PyTorch quantization. These results indicate that QDL serves as a generalized quantization simulator rather than a compressor-specific one, thereby conferring robustness across diverse quantization schemes.
>
> | Method           | Bit Acc.↑  | PSNR↑      | SSIM↑       | LPIPS↓       |
> |------------------|:----------:|:----------:|:-----------:|:------------:|
> | $Q_0=0.5$        | 94.27 | 27.15 | 0.843 | 0.195 |
> | $Q_0=1.5$        | 94.16 | 27.35 | 0.851 | 0.187 |
> | $Q_0=2.0$        | 94.29 | 27.12 | 0.842 | 0.195 |
> | $Q_0=1.0$ (Ours) | 94.03 | 27.55 | 0.844 | 0.173 |
>
> > **W5. Technically incorrect use of SSIM in Equation 5.**
>
> We thank the reviewer for pointing this out. We agree that SSIM is defined on local windows rather than individual pixels, and the current notation in Eq. 5 is too compact and can misleadingly suggest a single-pixel SSIM. In our implementation, however, we do not compute SSIM at a single pixel. For each position $p$, we take a local patch $N(I_hf, p)$, compute a window-based SSIM map
> $\mathrm{SSIM}_{win}(p) = \mathrm{SSIM}(N(I_hf, p), N(I'_hf, p))$
> and then define
>
> $P_{error}(p) = 1 - \mathrm{SSIM}_{win}(p)$.
>
> This error map is what we use in the high-frequency loss and in the median-based thresholding of FAG, so the implementation follows the standard patch-based definition of SSIM. In the revised version, we will update Eq. 5 to
> $P_{error}(p) = 1 - \mathrm{SSIM}_{win}(N(I_hf, p), N(I'_hf, p))$
> and add a short explanation to make it explicit that FAG is driven by a window-based SSIM error map rather than a single-pixel SSIM.
>
>
> > **W6.  The paper uses DWT for watermark extraction but switches to DFT for high-frequency detection without justification.**
>
> In our method, DWT and DFT play distinct yet complementary roles. DWT is applied **patch-wise**, and the decoder extracts the watermark solely from the LL band. **Since the decoder takes only the LL band as input, the gradients from the watermark loss backpropagate exclusively through these low-frequency components.** This naturally guides the watermark embedding to concentrate in the low-frequency domain, which is more robust against compression techniques like JPEG. In contrast, DFT is applied to both the rendered image and the ground truth to compute a **global high-frequency reconstruction error map**. FAG utilizes this signal to selectively place additional fine-scale Gaussians in high-frequency regions.
>
> Therefore, although all anchor features share the watermark via $f^w = f + \mathrm{tanh}(f')$, the terms **low-frequency embedding** and **high-frequency refinement** do not imply a strict frequency separation, but rather a **functional separation established during training**. Concretely, **the optimization focus for watermark recovery is directed toward the LL band, while FAG complements this by strengthening high-frequency details using small-scale, low-opacity Gaussians.** In the revised version, we will clarify this gradient-driven functional separation and the complementary use of DWT and DFT.
>
> [1] Training with Quantization Noise for Extreme Model Compression, ICLR 2021
>
> [2] UNIQ: Uniform Noise Injection for Non-Uniform Quantization of Neural Networks, TOCS 2021

---

> ### Author Response · Authors · 2025-11-24
>
> > **W7. Mask ambiguity and the limited effectiveness of $L_{hsv}$.**
>
> **1) The HSV loss penalizes all pixels in the range, including legitimate content.**
>
> In Eq. 7–8, the binary mask $M_c(p)$ only marks pixels in a given color range, but the HSV loss is always multiplied by $||I(p) - I_{gt}(p)||$. If a pixel is a genuine red pixel with $I(p) \approx I_{gt}(p)$, its contribution is nearly zero even when $M_c(p)=1$. **Thus, the loss does not penalize all pixels in that range uniformly, but focuses on pixels that both fall into the target range and deviate from the ground truth**. As described in Appendix C.6, we also threshold saturation and value so that low-saturation/low-value pixels are excluded, further concentrating the penalty on visible color artifacts.
>
> **2) Meaning of $|\Omega|$.**
>
> In Eq. 8, $\Omega$ denotes the full pixel domain of the image and $|\Omega|$ is the number of pixels used for normalization. Pixels with $M_c(p)=0$ contribute zero to the sum. We will explicitly state this in the revised paper.
>
> **3) Advantages and role of HSV loss.**
>
> HSV loss is a lightweight auxiliary term to suppress color artifacts introduced during watermark embedding. It converts RGB to HSV, masks only highly saturated and visually salient regions using H/S/V, and applies an MSE penalty on color differences there, making it much cheaper than perceptual losses such as LPIPS that require an additional deep network. As shown in the table (Mip-NeRF360, 48-bit), bit accuracy is almost identical, but HSV loss achieves higher PSNR/SSIM, while LPIPS loss provides only a small gain in the LPIPS score. Per-scene training time is also lower with HSV loss (**35.23** vs. **46.17** minutes). Thus, HSV loss is a simple, task-specific component that improves visual quality in artifact-prone regions while keeping bit accuracy similar and reducing computational overhead.
>
> | Method               | Bit Acc.↑ | PSNR↑   | SSIM↑     | LPIPS↓    | Training Speed↓ (Min.)   |
> |----------------------|:---------:|:-------:|:---------:|:---------:|:------------------------:|
> | Ours (w/ LPIPS loss) | 95.19 | 25.87 | 0.755 | **0.239** | 46.17 |
> | **Ours (w/ HSV loss)**   | **95.59** | **26.56** | **0.777** | 0.248 | **35.23** |
>
> >**W8. Limiting claims about generalizability across diverse compression paradigms.**
>
> To address the reviewer’s concern about generalization, we additionally evaluated the **SizeGS** compressor and observed that CompMarkGS remains more robust to compression than existing watermarking methods in this setting (Mip-NeRF 360, 48-bit). Appendix E.3 further shows that our method also outperforms prior work under **pruning-based compression** and **PyTorch float32$\to$int8 quantization**. These results confirm that QDL functions as a generalized quantization simulator rather than a compressor-specific one, effectively conferring robustness across diverse quantization mechanisms.
>
> The methods in [1,2] are full compression frameworks that re-encode vanilla 3DGS into their own compact representations, rather than plug-in modules that can be directly applied to an anchor-based 3DGS pipeline. Our work instead focuses on robustness to watermarking under quantization-based compressors built on anchor-based 3DGS, which is more memory-efficient and faster to render, and is increasingly adopted in recent high-performance 3DGS compression methods [3] (Appendix D.2).
>
> | Method                          | Bit Acc.↑      | PSNR↑          | SSIM↑          | LPIPS↓         |
> |---------------------------------|:-------------:|:-------------:|:-------------:|:-------------:|
> | WateRF             | 93.20 / 66.84 | 26.09 / 15.80 | 0.788 / 0.451 | 0.260 / 0.619 |
> | GaussianMarker     | 92.49 / 66.69 | 26.33 / 14.36 | 0.775 / 0.439 | 0.282 / 0.630 |
> | 3D-GSW             | 91.08 / 66.87 | 18.53 / 15.13 | 0.667 / 0.433 | 0.322 / 0.623 |
> | GuardSplat         | 71.33 / 64.46 | 16.15 / 16.45 | 0.524 / 0.462 | 0.483 / 0.612 |
> | CompMarkGS         | **95.59** / **83.79** | **26.56** / **23.59** | **0.777** / **0.673** | **0.248** / **0.351** |
>
> [1] Compact 3D Gaussian Representation for Radiance Field, CVPR 2024.
>
> [2] CompGS: Smaller and Faster Gaussian Splatting with Vector Quantization, ECCV 2024.
>
> [3] 3DGS.zip: A survey on 3D Gaussian Splatting Compression Methods, Eurographics 2025

---

> ### Author Response · Authors · 2025-11-24
>
> >**Q1. Mechanism and sensitivity of 3D-to-2D mapping.**
>
> We clarify the 3D Gaussian-to-2D high-frequency mapping and the FAG growing rule as follows.
>
> **1) 3D Gaussian to 2D high frequency mapping**
>
> For each view, we apply a high-pass filter to the rendered and ground-truth images, compute a window-based SSIM error map, and select pixels within a band around its median to form the high-frequency mask $I_{\mathrm{mask}}$. Each Gaussian is projected to $(u, v)$, rounded to the nearest pixel $(x, y)$, and labeled high-frequency if $I_{\mathrm{mask}}(x, y) = 1$.
>
> **2) When an anchor’s K Gaussians partially overlap high-frequency regions**
>
> If multiple Gaussians project to the same pixel, that pixel’s label is decided first and shared by all associated Gaussians. Anchor growing is then performed at the voxel level by aggregating gradients and opacity over Gaussians with $F_{\mathrm{mask}} = 1$, and creating a new anchor only when these statistics exceed a threshold and the same voxel consistently shows high-frequency error across views and iterations.
>
> **3) Coverage and sensitivity to the mapping heuristic**
>
> FAG constructs an SSIM-based high-frequency error map and, based on its median value $\tilde{P}_{error}$,
> selects only the pixels whose errors fall within the band
>
> $\tilde{P}_{error} \pm \epsilon$
>
> We also vary the masking threshold by setting $\epsilon \in {0.1, 0.3, 0.5}$ and evaluate performance on the Mip-NeRF 360 dataset under the 48-bit setting. Bit accuracy and PSNR/SSIM/LPIPS remain almost unchanged, indicating that FAG is not sensitive to the specific mapping heuristic. This is consistent with our design, where the gradient threshold in high-frequency regions mainly controls how many anchors are grown, and $\epsilon$ acts only as a secondary mask.
>
> | Method                  | Bit Acc.↑ | PSNR↑   | SSIM↑     | LPIPS↓    |
> |-------------------------|:---------:|:-------:|:---------:|:---------:|
> | $\epsilon\pm0.1$        | 95.57 | 26.58 | 0.7773 | 0.2480 |
> | $\epsilon\pm0.5$        | 95.40 | 26.58 | 0.7773 | 0.2482 |
> | $\epsilon\pm0.3$ (Ours) | 95.59 | 26.56 | 0.7770 | 0.2483 |

---

> > ### Comment · Reviewer_MFbA · 2025-11-26
> >
> > Thank you for the detailed responses. Most of my concerns have been addressed. Confirming that your method uses per-scene optimization with a single fixed message throughout training makes the experimental setup much clearer. The training cost breakdown also provides useful practical context.
> >
> > However, I still have one question regarding W6:
> >
> > Your explanation about "functional separation" rather than "strict frequency separation" actually contradicts Section 4.3's core motivation. The paper explicitly states there's a "fundamental conflict" when embedding watermarks and enhancing quality in the same low-frequency band. But if all anchors (including FAG-grown ones) embed watermarks via f^w = f + tanh(f_wf) and affect all frequency bands, how does FAG resolve this claimed conflict?

---

> ### Author Response · Authors · 2025-11-27
>
> We appreciate the reviewer's positive assessment and the clarification that most concerns have been addressed. Regarding the remaining question on W6, we clarify our intention below.
>
> In Sec. 4.3, our term "fundamental conflict" refers to an **optimization trade-off**, not a strict separation of frequency bands. In our training, we jointly optimize a **standard image reconstruction loss** $L_{scaffold}$ together with the **watermark loss** $L_{msg}$. Because the reconstruction loss is computed on the full rendered image, it affects all frequency components. In particular, the LL-band watermark signal is treated as an error relative to the ground-truth LL content. This creates a conflict between the watermark loss and the reconstruction loss. During optimization, watermark robustness tends to be weakened to improve image quality, or conversely, visual quality is sacrificed to preserve the watermark.
>
> FAG is introduced to mitigate this trade-off between rendering quality and watermark performance. Since the watermark loss is computed only from the DWT LL subband, watermark embedding mainly happens in the low-frequency band. In contrast, FAG uses a DFT-SSIM-based high-frequency error map and a median-band mask to add anchors only in regions with large high-frequency error, so that the additional updates are concentrated on repairing high-frequency reconstruction errors. Empirically, Figure 6 and Table 5 show that removing FAG consistently decreases both rendering quality and bit accuracy. This supports our view that FAG helps mitigate, rather than completely resolve, the conflict between LL watermark robustness and rendering quality.
>
> We will clarify this optimization perspective and the complementary roles of LL-band watermark embedding and FAG-based refinement in Sec. 4.3 of the revised manuscript.

---

> > ### Comment · Reviewer_MFbA · 2025-11-27
> >
> > I thank the authors for their thorough response. The clarification that Section 4.3 describes an optimization trade-off rather than strict frequency separation has resolved my concerns about W6. The empirical evidence from Figure 6 and Table 5, showing that FAG improves both rendering quality and bit accuracy, supports this optimization perspective. The explanation is now consistent with the methodology.
> >
> > I recommend that the authors include the training setup details from Round 1 (per-scene optimization and fixed watermark messages) in the revised manuscript.
> >
> > With these clarifications, I recommend acceptance of this paper. This work addresses a timely problem and will contribute valuable insights to the community on watermarking for emerging 3D representations. The authors' thorough responses have addressed my concerns and clarified the contributions. I will raise my score to 8 accordingly.

---

> > > ### Author Response · Authors · 2025-11-28
> > >
> > > We sincerely thank the reviewer for their continued engagement and for raising the score.
> > >
> > > We also appreciate the suggestion to include the training setup details from Round 1 (per-scene optimization and fixed watermark messages). We will incorporate these details into the revised manuscript to improve clarity.
> > >
> > > We deeply appreciate your constructive feedback throughout the review process and your recognition of our work's contribution to 3DGS watermarking.

---

### Official Review · Reviewer_n6dB · 2025-10-30

**Soundness:** 3
**Presentation:** 3
**Contribution:** 3
**Rating:** 6
**Confidence:** 4

**Summary:**

Targeting on copyright protection of 3D Gaussian Splatting (3DGS) models after compression, this paper proposes an anchor-based 3DGS watermarking framework, CompMarkGS, which directly adds learnable watermark-embedding features to the anchor features to enhance robustness against quantization. During training, a Quantization Distortion Layer (QDL) injects uniform noise with a learnable scale to explicitly simulate quantization rounding errors, aligning the model to be robust against post-compression. In parallel, a Frequency-Aware Anchor Growth (FAG) strategy adds anchors only in high-frequency error regions to balance visual quality, and an HSV loss is used to suppress color artifacts. Experimental results under anchor-based pipelines on Blender, LLFF, and Mip-NeRF360 evaluate performance before and after two mainstream quantization-based compressors (HAC, ContextGS), showing advantages over WateRF, GaussianMarker, 3D-GSW, and GuardSplat in both bit accuracy and visual quality; the trend holds across different message lengths and when swapping in an alternative decoder (self-supervised DINO). It is claimed that this is the first work to systematically address quantization-compression robustness for 3DGS watermarking with end-to-end validation.

**Strengths:**

The paper injects learnable quantization noise into anchor features via QDL, bringing the compressor’s distortion statistics forward into training, which is distinct from prior differentiable perturbation layers that only model image-domain distortions. By focusing squarely on robustness to quantization-based compression, the proposed FAG separates, in the frequency domain, the tension between visual enhancement (high frequencies) and watermark carrying (low-frequency LL), making apt use of anchor-based 3DGS structural priors. Overall, it is a new combination of structural choice + distortion modeling + frequency-aware growth.
The work offers comprehensive validation across three datasets and two compressors, evaluates both pre/post compression, covers image- and model-domain perturbations, varies message length, and tests multiple decoders; ablations show QDL/FAG/HSV each contributes substantially. After ContextGS/HAC compression, it outperforms strong baselines in bit accuracy and PSNR/SSIM.
In general, the method diagram and the training/extraction flow are explained clearly. As the anchor-based designs and compression are mainstream in the 3DGS ecosystem, addressing compression-induced watermark failure carries clear practical importance.

**Weaknesses:**

First, compression generalization could be strengthened. The current evaluation mainly covers HAC/ContextGS. The authors may consider adding different quantization bit-widths, per-channel/per-layer non-uniform quantization, and compressors with error feedback or post-entropy coding, to better match industrial diversity. Also, the QDL calibration strategy and mismatch-sensitivity analysis need to be reported for each setting.
Regarding baseline adaptation, GuardSplat should be moved from the SH space to an anchor-feature bias, which may introduce out-of-domain effects. It would help to report the native setting and a compatibility setting separately, or add a comparison axis of “native 3DGS + native compression”.
More targeted model-domain attacks and structural perturbations may be added to assess the limits of QDL/FAG under stronger adversaries.

**Questions:**

The frequency-aware anchor growth (FAG) sounds effective, but I’d like a more direct view of its costs and benefits. Could you add a page comparing model size, the number of anchors/Gaussians, and rendering latency (FPS or ms/frame) with and without FAG, and include a few figures showing where the added anchors are placed (aligned with your error/frequency heatmaps)? This would help assess its practical engineering overhead.

For “unknown/recompression” scenarios, double compression and bitrate switching across platforms are common in practice. I suggest adding a set of black-box re-compression experiments (e.g., compress with ContextGS then HAC, or multiple bitrate chains within the same compressor) to examine changes in bit accuracy and visual quality; also include stronger model-level attacks to help readers understand the method’s resilience under less friendly pipelines.

Regarding baselines, some methods need to be ported from their native representation space to the anchor-feature space. To avoid bias from implementation shifts, could you report the “native setting” (as faithful to the original paper as possible) and the “compatibility-modified” version separately, with key hyper-parameters and implementation details? This would make reproduction easier and fairness easier to judge.

---

> ### Author Response · Authors · 2025-11-25
>
> Thank you for your valuable comments and constructive feedback on our paper.
>
> > **W1. Compression generalization could be strengthened.**
>
> To address the concern about the diversity of compressors, we additionally evaluated CompMarkGS under **SizeGS** [1], a recently proposed anchor-based compressor. Crucially, SizeGS employs a particularly aggressive compression strategy that combines **per-channel mixed-precision quantization** with **pruning** to remove redundant anchors.
>
> This creates a dual challenge as quantization distorts the watermark signal, while pruning permanently erases the watermarked parameters themselves.
> Since SizeGS performs **50% pruning** followed by quantization, the performance degradation was slightly larger than with previous anchor-based 3DGS compression methods. However, as shown in the table (Mip-NeRF 360, 48-bit), CompMarkGS still achieved a high bit accuracy of **83.79%**. This is **17% higher** than the nearest baseline (3D-GSW), and our method also exhibits the smallest degradation in rendering quality. These results confirm that CompMarkGS generalizes well, maintaining robustness not only against standard quantization but also against heterogeneous compressors that involve both mixed-precision distortion and structural pruning."
>
> | Method         | Bit Acc.↑      | PSNR↑        | SSIM↑         | LPIPS↓        |
> |----------------|:-------------:|:-------------:|:-------------:|:-------------:|
> | WateRF         | 93.20 / 66.84 | 26.09 / 15.80 | 0.788 / 0.451 | 0.260 / 0.619 |
> | GaussianMarker | 92.49 / 66.69 | 26.33 / 14.36 | 0.775 / 0.439 | 0.282 / 0.630 |
> | 3D-GSW         | 91.08 / 66.87 | 18.53 / 15.13 | 0.667 / 0.433 | 0.322 / 0.623 |
> | GuardSplat     | 71.33 / 64.46 | 16.15 / 16.45 | 0.524 / 0.462 | 0.483 / 0.612 |
> | CompMarkGS     | **95.59** / **83.79** | **26.56** / **23.59** | **0.777** / **0.673** | **0.248** / **0.351** |
>
> > **W2. The QDL calibration strategy and mismatch-sensitivity analysis need to be reported for each setting.**
>
> **1) QDL calibration strategy.**
>
> QDL is not manually calibrated for each compression method. We use a single global configuration and, as described in Sec. 4.1 and 5.1, we fix the base scale $Q_0 = 1.0$ for all experiments. A small network  $\mathrm{MLP}_q$ takes the watermarked anchor feature $f_i^w$ as input, predicts a refinement $r_i$, and outputs the final quantization scale $q_i = Q_0 \cdot (1 + \tanh(r_i))$. This keeps $q_i$ in the bounded range $(0, 2Q_0)$, preventing the QDL noise from collapsing to zero or becoming arbitrarily large. As a result, we use the same QDL architecture and hyperparameters across all settings (HAC, ContextGS, PyTorch int8 quantization), without any additional per-scene or per-compressor calibration.
>
> **2.) Mismatch sensitivity.**
>
> We additionally vary the QDL base scale $Q_0$ under the ContextGS compressor with a 48-bit setting, using $Q_0\sim{0.5, 1.0, 1.5, 2.0}$. As summarized in the table below, the post-compression bit accuracy remains in a very narrow range, and PSNR/SSIM also change only slightly. This shows that QDL is not overly sensitive to moderate miscalibration of the noise magnitude.
>
> To evaluate a stronger cross-pipeline mismatch, Appendix E.3 reports experiments under a float32→int8 quantization scheme (PyTorch quantization), which differs significantly from the anchor-based 3DGS compression methods we target. In this setting, existing 3DGS watermarking methods lose more than **20\%** in bit accuracy, whereas CompMarkGS with QDL drops by only about **5\%**, showing that our method remains relatively robust even when the quantization used at test time differs from the one used during training. As discussed in W1, we also evaluate the SizeGS compressor and still observe over **80\%** bit accuracy, further indicating that our robustness is not tied to a single quantization pipeline.
>
> | Method           | Bit Acc.↑  | PSNR↑      | SSIM↑       | LPIPS↓       |
> |------------------|:----------:|:----------:|:-----------:|:------------:|
> | $Q_0=0.5$        | 94.27 | 27.15 | 0.843 | 0.195 |
> | $Q_0=1.5$        | 94.16 | 27.35 | 0.851 | 0.187 |
> | $Q_0=2.0$        | 94.29 | 27.12 | 0.842 | 0.195 |
> | $Q_0=1.0$ (Ours) | 94.03 | 27.55 | 0.844 | 0.173 |
>
> [1] SizeGS: Size-aware Compression of 3D Gaussians with Hierarchical Mixed Precision Quantization, MM 2025

---

> ### Author Response · Authors · 2025-11-25
>
> > **W3&Q3. Regarding baselines, some methods need to be ported from their native representation space to the anchor-feature space.**
>
> GuardSplat was originally designed for native 3DGS, where the **watermark offset** is added to the **SH coefficients** that control Gaussian color. In native 3DGS, SH coefficients play the same role that **anchor features** passed through an MLP play in anchor-based 3DGS. Since SH parameters are not available in our anchor-based 3DGS and compression setting, we adapt GuardSplat by adding the watermark offset to the anchor features, preserving its idea of perturbing the color-controlling representation. We use this implementation as a compatibility baseline in the main paper and Appendix D.1.
>
> For a fair comparison where all methods share the same anchor-based backbone and 48-bit compression setting, the main tables report the performance of this compatibility baseline. Following the reviewer’s suggestion, we also report GuardSplat results for both the original native-3DGS setting and the compatibility setting under the same pruning ratio (**20%**) in the 48-bit configuration, and clarify in the revised version which numbers correspond to each setting.
>
> | Method              | Bit Acc.↑     | PSNR↑         | SSIM↑         | LPIPS↓        |
> |---------------------|:-------------:|:-------------:|:-------------:|:-------------:|
> | GuardSplat (anchor) | 76.49 / 59.94 | 16.60 / 12.10 | 0.619 / 0.344 | 0.385 / 0.459 |
> | GuardSplat (native) | 94.22 / 74.37 | 38.15 / 26.64 | 0.991 / 0.928 | 0.005 / 0.053 |
>
> > **W4. More targeted model-domain attacks and structural perturbations may be added to assess the limits of QDL/FAG under stronger adversaries.**
>
> We additionally evaluated **post-training quantization (PTQ)** on internal MLP weights. As shown in the table (Blender, LLFF, MipNeRF-360 / 48-bit), CompMarkGS remains robust under MLP PTQ. Combined with the pruning results in Appendix E.3, this confirms our method's stability against structural model changes. Furthermore, under **PGD attacks** targeting the decoder (Appendix E.4), our model maintained **84.33\%** accuracy, significantly outperforming baselines that remained around **50\%**.
>
> Regarding limitations, we observed that bit accuracy degrades when HAC, ContextGS, and pruning intensities are pushed to extreme limits (Appendix E.6, 7). However, this degradation occurs only in regimes where rendering quality has already collapsed to levels unacceptable for practical use. Our method explicitly targets quantization distortions in realistic compression scenarios, rather than claiming a universal defense against arbitrary attacks.
>
> | Method              | Bit Acc. (None)↑  | Bit Acc. (PTQ)↑  |
> |---------------------|:-----------------:|:----------------:|
> | CompMarkGS          | 95.16 / 94.98   | 87.78  / 70.28  |

---

> ### Author Response · Authors · 2025-11-25
>
> > **Q1. Could you add a page comparing model size, the number of anchors/Gaussians, and rendering latency with and without FAG, and include a few figures showing where the added anchors are placed?**
>
> In response to the reviewer’s request, we report an overhead comparison with and without FAG on the Mip-NeRF 360 dataset. As shown in the table below, applying FAG results in a moderate increase in model size and the number of anchors. However, the training time increases only marginally by approximately **3\%**.
>
> Regarding rendering latency, although the speed decreases from **62.27 FPS** to **57.34 FPS**, it remains well above the **30 FPS** threshold, robustly guaranteeing real-time performance. Furthermore, as shown in Table 14 of Appendix F, even with FAG enabled, our method is still faster than other watermarking baselines such as 3D-GSW and WateRF.
>
> We have updated **Appendix G** with visualizations to verify the efficacy of FAG. The FAG mask serves an auxiliary role by defining spatial candidates for the anchor growth, while the accumulated gradient threshold determines whether an anchor is actually generated. **Figure 11** visualizes an **intermediate state during the training phase**, showing how **new anchors (FAG)** are selectively generated within the **high-frequency SSIM-based error map**. Specifically, anchors emerge in regions where high-frequency errors have consistently accumulated to trigger the gradient threshold. This demonstrates that FAG does not indiscriminately densify all high-frequency areas but effectively targets regions requiring structural refinement.
>
> | Method               | Size↓ (MB)  | Number of anchors (K) | FPS↑          |Training Speed↓ (Min.) |
> |----------------------|:-----------------:|:---------------------:|:-------------:|:---------------------:|
> | CompMarkGS (w/o FAG) | 319.79 | 766.4 | 62.27 | 35.26 |
> | CompMarkGS (w/ FAG)  | 377.02 | 906.6 | 57.34 | 36.36 |
>
> >**Q2. Adding a set of black-box re-compression experiments to examine changes in bit accuracy and visual quality.**
>
> Following the reviewer’s suggestion, we additionally ran experiments with multiple-bitrate chains within the same compressor. Starting from a model trained and compressed on MipNeRF360 at $\lambda{=}0.004$, we re-compressed it twice at lower bitrates. As the model size decreased from **14.95 MB to 11.95 MB**, the watermark bit accuracy stayed almost the same, whereas PSNR/SSIM gradually dropped. This suggests that re-compression mainly injects noise into high-frequency details and overall pixel values, which first harms standard image quality metrics, whereas our method recovers the watermark from a DWT-based low-frequency representation. As a result, the watermark signal remains stable even after multiple bitrate changes and repeated recompressions.
>
> Also, our current evaluation already includes a related two-stage pipeline. In Appendix E.3, **we first compress the anchor-based 3DGS with HAC or ContextGS and then apply LightGaussian**, and CompMarkGS consistently maintains higher bit accuracy than prior 3DGS watermarking methods at comparable PSNR/SSIM.
>
> | Stage | Size↓ (MB)  | Bit Acc. ↑ | PSNR ↑ | SSIM ↑ | LPIPS ↓ |
> | :---: | :---: | :---: | :---: | :---: | :---: |
> | 0 | 14.95 | 95.48 | 26.57 | 0.7770 | 0.2480 |
> | 1 | 13.45 | 95.36 | 25.64 | 0.7308 | 0.2620 |
> | 2 | 11.95 | 95.16 | 24.90 | 0.6965 | 0.2743 |

---

### Official Review · Reviewer_YKCM · 2025-10-31

**Soundness:** 2
**Presentation:** 2
**Contribution:** 2
**Rating:** 6
**Confidence:** 4

**Summary:**

This paper presents a novel compression-tolerant anchor-based framework for 3DGS watermarking that protects the ownership of 3DGS model. The authors focus on preserving watermark integrity and rendering quality effectively both before and after compression. The framework is evaluated on multiple datasets, demonstrating that it can achieve state-of-the-art performance.

**Strengths:**

1.	A compression-tolerant anchor-based watermarking method is proposed to embed a learnable watermark embedding feature into the anchor feature, which improves robustness against model compression.
2.	The authors perform experiments on different datasets and investigate robustness by measuring bit accuracy under various distortions before and after compression.

**Weaknesses:**

1.	What is the true innovation of the quantization distortion layer? Is it merely a transfer of technique from the 3DGS compression method?
2.	What is the true advantage of HSV loss? Can you compare HSV loss with the effects of using perceptual loss?
3.	The experiments were primarily validated on anchor-based 3DGS. Does it exhibit the same performance for standard 3DGS?
4.	In Table 2, for model distortion robustness, how does the method perform under additional 3D geometric attacks, such as 3D translation, rotation, or crop-out (which is evaluated in GaussianMarker) ?
5.	The text in Figure 2 may need to be revised for clearer expression.

**Questions:**

See the weakness

---

> ### Author Response · Authors · 2025-11-24
>
> Thank you for your valuable comments and constructive feedback on our paper.
>
> > **W1. What is the true innovation of the quantization distortion layer? Is it merely a transfer of technique from the 3DGS compression method?**
>
> The **Quantization Distortion Layer (QDL)** is a module tailored to the quantization mechanism used in anchor-based 3DGS. Although inspired by quantization-based 3DGS compression methods, **our contribution is to introduce this quantization process into the watermarking pipeline as a differentiable proxy**. Prior 3DGS watermarking works mainly consider traditional distortions such as JPEG and resizing, and do not directly address the practical issue of degraded watermark recoverability caused by model compression.
>
> Specifically, QDL is applied to the watermarked anchor feature $f^w$ with a learnable per-anchor feature noise scale. For each $f^w_i$, we sample $\varepsilon_i \sim \mathcal{U}(-0.5, 0.5)$ and define $\tilde{f}^w_i = f^w_i+q_i\odot\varepsilon_i$ (Eq. (4)). During backpropagation, $\epsilon_i$ is regarded a constant, so $\tilde{f}^w_i$ is affine and hence differentiable. In contrast to the non-differentiable rounding operation used in actual quantization, this additive noise formulation provides a smooth surrogate that mimics quantization-induced perturbations while still allowing end-to-end training that preserves both bit accuracy and rendering quality after compression.
>
> As shown in Table 5, removing QDL under strong quantization sharply degrades bit accuracy, whereas with QDL, our method consistently maintains over **90\%** accuracy across HAC, ContextGS, and PyTorch quantization, supporting QDL as an effective proxy for anchor space quantization in the watermarking scenario.
>
> > **W2. What is the true advantage of HSV loss? Can you compare HSV loss with the effects of using perceptual loss?**
>
> As discussed in **Sec. 4.1** and **5.3**, we propose embedding the watermark into the anchor feature rather than position or scaling to minimize structural distortions. However, since anchor features are involved in predicting Gaussian colors, embedding information here tends to induce **local color artifacts**.
>
> While Perceptual Loss (LPIPS) excels at preserving global structure and texture, it tends to overlook these subtle chromatic shifts when the underlying structure remains intact. In contrast, **our proposed HSV loss explicitly masks highly saturated and visually salient regions, applying a concentrated penalty to color discrepancies in those areas**.
>
> As shown in the table below (Mip-NeRF 360, 48-bit), while using LPIPS loss naturally resulted in a slightly better LPIPS score, HSV loss achieved superior PSNR, SSIM, and training speeds. For these reasons, we propose HSV loss as a practical choice that secures both rendering fidelity and training efficiency.
>
> | Method               | Bit Acc.↑ | PSNR↑   | SSIM↑     | LPIPS↓    | Training Speed↓ (Min.)   |
> |----------------------|:---------:|:-------:|:---------:|:---------:|:------------------------:|
> | Ours (w/ LPIPS loss) | 95.19 | 25.87 | 0.755 | **0.239** | 46.17 |
> | **Ours (w/ HSV loss)**   | **95.59** | **26.56** | **0.777** | 0.248 | **35.23** |

---

> ### Author Response · Authors · 2025-11-24
>
> > **W3. The experiments were primarily validated on anchor-based 3DGS. Does it exhibit the same performance for standard 3DGS?**
>
> As discussed in **Appendix D.2**, anchor-based 3DGS provides better memory and rendering efficiency than standard 3DGS. Also, recent state-of-the-art compression methods predominantly adopt anchor-based representations [1]. Consequently, we strategically selected anchor-based 3DGS.
>
> **From a technical perspective, the core components of CompMarkGS (QDL, FAG, and HSV Loss) are conceptually architecture-agnostic**. Since QDL injects quantization noise into parameter vectors and both FAG and HSV Loss operate on rendered images, they could, in principle, be applied to standard 3DGS.
> However, given that compression pipelines for standard 3DGS differ significantly from those for anchor-based models, we prioritized validating our method within the anchor-based compression scenario, which represents the current frontier of 3DGS compression performance. We consider the systematic extension to standard 3DGS an intriguing and important direction for future work.
>
> [1] 3DGS.zip: A survey on 3D Gaussian Splatting Compression Methods, Eurographics 2025

---

> ### Author Response · Authors · 2025-11-24
>
> > **W4. How does the method perform under additional 3D geometric attacks?**
>
> Following the reviewer's suggestion, we conducted additional evaluations on the 3D geometric attacks presented in GaussianMarker. Among the suggested attacks, **Gaussian noise** was already analyzed in our main paper. Regarding **Crop-out**, we consider our robustness already demonstrated via the **pruning attack** in Table 2, as both attacks share the fundamental mechanism of data loss. Therefore, we focused our additional experiments on **Rotation and Translation**, which were not previously covered.
>
> As shown in the table below (Blender, LLFF, Mip-NeRF 360 / 48-bit), while geometric distortions inevitably degrade performance across all methods, CompMarkGS demonstrates significantly higher resilience, maintaining a smaller drop in bit accuracy compared to baselines.
> This robustness is due to our embedding strategy. By using the anchor feature level instead of changing geometric attributes, the signal remains stable against such distortions.
>
> | Method         | Bit Acc. (None) | Bit Acc. (Rot. $r=\pm\pi/6$) | Bit Acc. (Trans. $t=[0, 1000]^3$) |
> |----------------|:-------------:|:-------------:|:-------------:|
> | WateRF         | 91.52 / 72.38 | 53.75 / 53.96 | 53.75 / 53.96 |
> | GaussianMarker | 91.62 / 73.15 | 57.99 / 57.96 | 55.46 / 55.57 |
> | 3D-GSW         | 89.62 / 62.89 | 53.50 / 53.38 | 53.50 / 53.38 |
> | GuardSplat     | 76.49 / 59.94 | 59.15 / 55.87 | 59.15 / 55.87 |
> | **CompMarkGS**     | **95.16** / **94.98** | **73.92** / **70.62** | **87.76** / **70.28** |
>
> > **W5. The text in Figure 2 may need to be revised for clearer expression.**
>
> Thank you for pointing out that the text description in **Figure 2** could be improved. In the revised manuscript, we have rewritten the caption to clearly distinguish the watermark embedding feature $f'$ from the watermarked anchor feature $f^w$ and to provide a more precise description of the pipeline flow. We believe this textual revision resolves the ambiguity and makes the overall method easier to understand.

---

### Comment · Area_Chair_GzaT · 2025-11-25

Dear Reviewers,

Thank you for your time and effort in reviewing submissions for ICLR 2026. As we begin the author-reviewer discussion process, we kindly remind you to submit your responses to the author rebuttals by **December 2**.

Your engagement in this discussion phase is crucial to ensuring a fair and thorough evaluation of each submission.

### **Action Required**
- Carefully consider the authors’ rebuttal and any additional evidence they provide.
- Update your review (if applicable) to reflect your revised perspective.
- Discuss with the authors if further details are required

Your AC

---

### Author Response · Authors · 2025-12-03
**Rebuttal Summary**

We are deeply grateful for the time and care you have devoted to reviewing our submission. Below, we highlight the key contributions of our paper and summarize the main points discussed during the rebuttal period.

## **Key Contributions**
1) **Compression-tolerant 3DGS watermarking framework**: We introduce an anchor-based 3DGS watermarking framework to remain robust after quantization-based compression. This addresses an important practical problem that previous 3DGS watermarking methods have mostly ignored, which is watermark failure after compression.

2) **Anchor-feature watermark embedding**: Instead of directly embedding geometric attributes of Gaussians, we embed a learnable watermark embedding feature into anchor features.

3) **Quantization distortion layer(QDL)**: We use a QDL that injects uniform noise with a learnable noise scale into the watermarked features during training. This acts as a quantization-aware surrogate and exposes the encoder to quantization-induced distortions.

4) **Frequency-aware anchor growing(FAG) & HSV loss**: To improve rendering quality, we propose FAG, which selectively densifies anchors in high-frequency error regions guided by a SSIM-based error map. We also introduce HSV loss that suppresses visible color artifacts in highly saturated regions.

## **Summary of Rebuttal Discussions**

The reviewers highlighted the following strengths of our work:

* __Motivation__: Reviewers **n6dB, MFbA, tMZM** agreed that addressing robustness to quantization-based compression is a timely and valuable contribution. In particular, reviewer **n6dB** emphasized that, as the anchor-based designs and compression are mainstream in the 3DGS ecosystem, addressing compression-induced watermark failure carries clear practical importance.

* __Contribution__: Overall, all reviewers found that our method is technically sound and makes a meaningful contribution in the context of 3DGS watermarking. Reviewer **n6dB**, in particular, noted that CompMarkGS presents a new combination of structural choice + distortion modeling + frequency-aware growth.

* __Comprehensive validation__: Reviewers **YKCM, n6dB, and MFbA** noted that our broad experimental evaluation provides solid support for the robustness and practical relevance of our method.

In addition, during the discussion we addressed the reviewers' main concerns as follows:

1) **Clarification of key modules**

* **QDL**: We clarified its configuration, role, and effectiveness in more detail. And through additional experiments on varying the noise scale, we confirmed that the watermarking performance remains stable under quantization-induced distortion.

* **FAG**: We corrected the notation to make it explicit that a window-based SSIM error map is used. We revised Sec. 4.3 so that the term "fundamental conflict" is framed as an optimization trade-off between watermark robustness and rendering quality, rather than implying a strict separation of frequency bands.

* **HSV loss**: We clarified that it focuses on highly saturated and visually salient pixels that deviate from the ground truth, thereby suppressing color artifacts. We also showed that compared to using a perceptual loss (LPIPS), HSV loss improves PSNR/SSIM and reduces training time while maintaining comparable bit accuracy.

2) **Experimental reinforcement**

* **Compression generalization**: We added experiments with **SizeGS** and **multi-bitrate recompression within the same compressor**, strengthening our claims about generalization across diverse compression schemes.

* **Model distortion**: We also included **MLP PTQ** and **geometric attacks**, further confirming the robustness of our method to model distortions.

* **Decoder dependence**: We added new experiments with the **3D decoder**. And we revisited results already reported in the Appendix for an **SSL-based decoder** and **PGD adversarial attacks that target the decoder**. Taken together, these results clarify that our method is not tied to a specific decoder and substantially alleviate concerns about vulnerability to decoder-targeted adversarial attacks.

3) **Overhead**

* We reported training time, memory usage, and FPS relative to the Scaffold-GS backbone, demonstrating that the additional overhead introduced by the watermarking modules (QDL, FAG, HSV) is limited.

* We further compared model size, the number of anchors, FPS, and training time with and without FAG, showing that FAG increases training time only slightly, **while still maintaining real-time rendering (>30 FPS) and remaining faster than baselines**.

---

### Author Response · Authors · 2025-12-03
**Final Remarks**

Dear Area Chair and Reviewers,

We are deeply grateful for the time and care you have devoted to reviewing our submission.

Before the discussion, our paper received scores of **(6, 6, 2, 6)**. After the rebuttal and subsequent clarification, the scores were updated to **(6, 6, 8, 6)**.

During the discussion, reviewers agreed on the importance of robustness to quantization-based compression, appreciated the effectiveness of our key modules, and recognized the scope of our experiments.
For the main concerns raised during the discussion, we clarified the training protocol and key components and added experiments across diverse compressors, model distortions, and decoders to better support our claims on generalization and robustness. In addition, we quantified the overhead of FAG and confirmed that the overhead is limited while preserving real-time performance.

We believe that the main concerns raised during the discussion have been substantially addressed through additional experiments and clarified exposition. In particular, reviewer **MFbA** initially gave a score of **2**, but after reconsidering the paper in light of our clarifications, raised it to **8**. In their final comment, **MFbA** not only noted that our work tackles the timely problem of "3DGS watermarking under compression" and provides meaningful insights to the community, but also explicitly stated that they would raise their rating to 8. We regard this as strong evidence that our compression-aware design and additional experiments have effectively addressed the main concerns.

Furthermore, all suggestions made by reviewers **n6dB and MFbA** have been incorporated into the revised manuscript. All modified or newly added parts are highlighted in **"blue"** for ease of inspection by the Area Chair and reviewers.

We sincerely appreciate the time and valuable feedback provided by you and the reviewers, and we hope you will consider these revisions and expanded results in your final recommendation. We believe that the paper now provides a meaningful and practically relevant contribution to watermarking for compressed 3DGS assets.

---

### Meta-Review · Area_Chair_18H6 · 2026-01-15

**Summary:**

[AC: Overall, the reviewers are positive toward this work. I do not see any ill intent during the rebuttal.]

Reviewer YKCM (6: marginally above the acceptance threshold. But would not mind if paper is rejected; 4: You are confident in your assessment)
[AC: Most comments are clarification ones, and are addressed with more results.]

Reviewer n6dB (6: marginally above the acceptance threshold. But would not mind if paper is rejected; 4: You are confident in your assessment)

[AC: Most comments are addressed with more results, confirming the superiority of the proposed method.]

Reviewer n6dB (2: reject, not good enough; 3: You are fairly confident in your assessment.)

[AC: This reviewer has the lowest confidence and asked several questions. During the rebuttal period, he asked one more question but finally indicated that he was pleased with the responses from the authors, and planned to raise his score to 8.]

**Reviewer Concerns:**

I believe most comments have been addressed properly. This paper is good to be considered for acceptance.

**Reviewer Scores:**

The initial ratings are 6, 6, 2. The reviewer with the lowest score was convinced by the authors' responses. Overall, the reviewers are positive toward this work. I do not see any ill intent during the rebuttal.

---

### Decision · Program_Chairs · 2026-01-26

Accept (Poster)